# Enhanced Self-Distillation Framework for Efficient Spiking Neural Network Training

**Xiaochen Zhao**[1†] **Chengting Yu**[1,2†] **Kairong Yu**[1,2] **Lei Liu**[1] **Aili Wang**[1,2*]

[1] ZJU-UIUC Institute, Zhejiang University
[2] College of Information Science and Electronic Engineering, Zhejiang University
{xiaochen.24,chengting.21,ailiwang}@intl.zju.edu.cn

## Abstract

Spiking Neural Networks (SNNs) exhibit exceptional energy efficiency on neuromorphic hardware due to their sparse activation patterns. However, conventional training methods based on surrogate gradients and Backpropagation Through Time (BPTT) not only lag behind Artificial Neural Networks (ANNs) in performance, but also incur significant computational and memory overheads that grow linearly with the temporal dimension. To enable high-performance SNN training under limited computational resources, we propose an enhanced self-distillation framework, jointly optimized with rate-based backpropagation. Specifically, the firing rates of intermediate SNN layers are projected onto lightweight ANN branches, and high-quality knowledge generated by the model itself is used to optimize substructures through the ANN pathways. Unlike traditional self-distillation paradigms, we observe that low-quality self-generated knowledge may hinder convergence. To address this, we decouple the teacher signal into reliable and unreliable components, ensuring that only reliable knowledge is used to guide the optimization of the model. Extensive experiments on CIFAR-10, CIFAR-100, CIFAR10-DVS, and ImageNet demonstrate that our method reduces training complexity while achieving high-performance SNN training. Our code is available at https://github.com/Intelli-Chip-Lab/enhanced-self-distillation-framework-for-snn.

## 1 Intorduction

Spiking Neural Networks, which emulate the dynamic behavior of biological neurons [52], transmit information between neurons through binary spike events across synapses [47, 56]. This spike-based architectural design enables remarkable energy efficiency on neuromorphic hardware [1, 8, 54]. However, mainstream training for SNNs relies on time-unfolded chain rules and Backpropagation Through Time (BPTT). Due to the intrinsic temporal dynamics of SNNs, training with BPTT requires storing the entire computational graph across all time steps, resulting in significant memory and computational overhead [42, 83, 35, 69, 68]. To address this issue, recent studies have begun focusing on improving the training efficiency of SNNs by decoupling the chain rule, aiming to reduce the complexity of BPTT [2, 48, 49, 69, 3, 86, 77, 48]. These approaches significantly reduce both time and memory consumption while achieving performance comparable to BPTT [86, 77]. However, the simplifications introduced in the gradient computation graph often lead to gradient distortions in intermediate layers, which tend to accumulate and become more pronounced as the network depth increases [77]. Recent works have SNNs exhibit feature representations based on firing rates that closely resemble those of artificial neural networks (ANNs) [44, 4, 72]. We adhere to the rate-coding assumption [77], treating the spiking rate as the fundamental unit of training and designing dedicated strategies specifically tailored to it. This work proposes introducing

---

[†]Equal contribution [*]Corresponding author

39th Conference on Neural Information Processing Systems (NeurIPS 2025).

lightweight auxiliary ANN branches to map the firing rate from intermediate SNN layers through the corresponding auxiliary ANN branches and guiding them with additional supervision signals. The auxiliary branches are supposed to backpropagate more accurate gradients to the associated substructures and intermediate implicit rate representations, mitigating the gradient distortion issues inherent in efficient training frameworks. Importantly, the backpropagation in branches operates solely on rate-based activations during both forward and backward passes, making it highly compatible with the rate-based framework and preserving its advantages in training efficiency on time and memory overhead. Moreover, as auxiliary branches are discarded during inference, our method incurs no additional computational cost at inference time. A quick comparison is shown in Fig. 1, illustrating the positioning of our method within the landscape of direct SNN training schemes.

Knowledge Distillation (KD) is an effective strategy for enhancing model performance by leveraging soft labels provided by a teacher model [31, 84] and has recently been applied in SNNs training [22, 72, 71, 32, 78]. Building on logits-based KD, the self-distillation framework [82] was proposed to leverage the model's own knowledge to guide its learning process, thereby eliminating the need for an additional pre-trained teacher model. The idea behind self-distillation is that the final layer, benefiting from the entire network, can produce higher-quality predictions, which can be used as teacher labels to optimize the learning of intermediate layers [82, 12]. However, in the self-distillation framework for SNNs, we observe that the teacher labels are inherently associated with different stages of the model's training process[74], and their convergence rates may vary

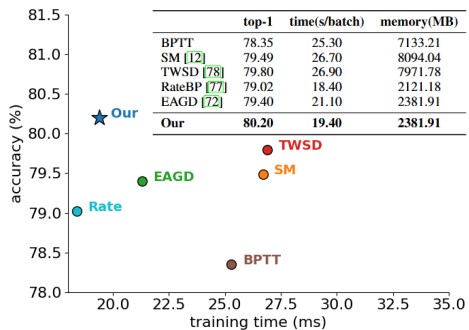

Figure 1: The performance of different training methods on ResNet-18 with the CIFAR-100 dataset shows that our approach achieves superior training efficiency and accuracy compared to current mainstream methods.

due to the varying structural complexities across layers. We introduce the concept of label reliability to measure the ability of a teacher signal to effectively guide the student model toward correct convergence. Our observations indicate that the final layer's output is not always the most reliable throughout training, and statically assigning it as the teacher may lead the model to converge to a suboptimal solution. To address this issue, we propose an enhanced self-distillation framework that disentangles the reliable and unreliable components from the predictions of multiple branches, aggregating only the reliable parts to form teacher labels. The framework enables the model to iteratively leverage its own reliable knowledge while mitigating the negative impact of unreliable outputs on the student model, thereby improving the effectiveness of self-distillation in optimizing the backbone network. Our contributions can be summarized as follows:

- We establish a mapping between the firing rates of intermediate layers and the ANN branch, optimizing the gradient errors of the intermediate layers under rate-based backpropagation. This results in outstanding performance with extremely low training cost.

- We analyze the reliability of teacher signals in the self-distillation process and propose a novel decoupling strategy that separates reliable and unreliable components, thereby constructing a more stable and effective self-distillation framework.

- We conduct empirical validation on standard datasets, including CIFAR-10, CIFAR-100, CIFAR10-DVS, and ImageNet, and perform ablation studies on various model components. Our results demonstrate that the proposed framework effectively balances training efficiency and high performance, offering clear advantages over existing methods.

## 2  Related Work

### 2.1  Training Methods in SNNs

The primary training approaches for Spiking Neural Networks (SNNs) include conversion-based methods and direct training methods. The former establishes a connection between SNNs and Artificial Neural Networks (ANNs) via an equivalent closed-form mapping, converting a pre-trained ANN into the target SNN model. This avoids the challenge of training SNNs from scratch. Although

several studies [10, 26, 27, 41] have achieved nearly lossless accuracy through such conversions [6, 13, 28, 59, 57, 10, 41, 14], the resulting SNNs typically require longer inference times to match the accuracy of the original ANN [5, 39, 30, 29, 34]. In contrast, direct training methods compute gradients of discrete spikes using surrogate gradients and Backpropagation Through Time (BPTT), allowing SNNs to achieve competitive performance with very few time steps [51, 60, 67, 20, 75, 85, 80, 42, 61, 66, 83, 73]. However, since the training methods rely on temporal backpropagation, they introduce both time and memory overhead that scales linearly with the number of time steps, as the computational graph must be retained during training [42, 35, 68, 69, 48, 12]. Recent works have proposed methods to decouple the temporal dimension of the backpropagation computation graph [50, 55, 65, 79, 2, 3, 76, 69, 48, 86]. Among them, [77] proposed a targeted training framework for rate-based representations, simplifying the cost of backpropagation-based training significantly. However, efficient training schemes require temporal approximations, which consequently introduce additional gradient errors that accumulate as the network depth increases.

## 2.2 Knowledge Distillation in SNNs

Knowledge distillation is a common transfer learning technique that facilitates the transfer of knowledge from a high-capacity model to a lower-capacity one [31, 45, 62, 64, 33, 21], enabling a smaller student model to approximate the performance of a larger teacher model. In the context of Spiking Neural Networks, a typical strategy involves using a pre-trained ANN or a larger SNNs to guide the training of a smaller SNN [37, 38, 63, 81, 70, 24]. Among them, KDSNN [71] adopts a joint distillation method based on both logits and feature representations, while [32] introduces a hierarchical feature distillation framework. Although these methods have demonstrated the effectiveness of knowledge distillation for SNNs across various datasets, they share a common limitation: the requirement for additional training on larger networks, which incurs significant computational overhead. [82, 12] proposed a self-distillation strategy that leverages the model's own knowledge to guide the learning of different parts of the network. However, in the process of self-distillation, the quality of teacher labels is coupled with the training stage of the model. In the early training phase, statically assigned teacher labels may be unreliable, potentially misleading the student model. To address this issue, studies such as [74] have proposed normalized distillation losses and customized soft label schemes, which improve the quality of teacher labels by smoothing the target class probabilities and adjusting the distribution over non-target classes. However, when high-confidence predictions correspond to incorrect classes, the teacher labels may exert an even stronger misleading effect on the student.

## 3 Method

### 3.1 Preliminary

Inspired by the mechanism of discrete pulse transmission in the brain, Spiking Neural Networks (SNNs) utilize spiking neurons as their basic computational units. Among these models, the Leaky Integrate-and-Fire (LIF) neuron is the most commonly used in SNN training. In this model, when the inputs received by a neuron raise its membrane potential to a specific threshold, a binary pulse is triggered, and subsequently, the membrane potential is reset. The equations for the membrane potential dynamics and spike generation are described as follows:

$$V^l\left[t+1\right] = \lambda \cdot \left(V^l\left[t\right] - V_{th} \cdot S^{l-1}\left[t\right]\right) + W^l \cdot S^l\left[t\right] + b^l \tag{1}$$

$$S^l\left[t+1\right] = H\left(V^l\left[t+1\right] - V_{th}\right) \tag{2}$$

Here, $\lambda$ represents the membrane time constant, $V_{th}$ is the membrane threshold, $W^l$ and $b^l$ denote the weights of the neurons in the $l$-th layer. $V^l[t]$ and $S^l[t]$ represent the membrane potential and spike emission of the neurons in the $l$-th layer at time $t$, respectively. $H(\cdot)$ is the step function that generates spikes, and due to its non-differentiability, gradient-based methods are used during backpropagation to propagate the error.

### 3.2 Model training

Distinguished from conventional BPTT, which performs a single forward and backward propagation, our training procedure is divided into two distinct stages (in Figure 2). In the first stage, a forward

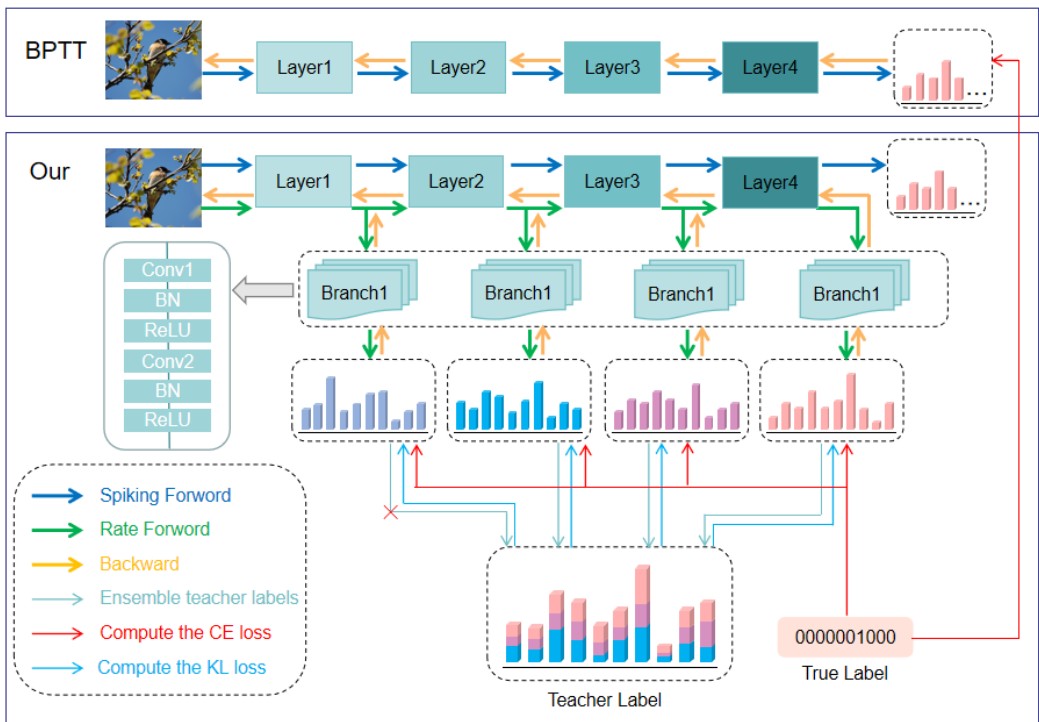

Figure 2: Framework Overview. Unlike the standard BPTT approach, we propose a rate-based framework that first performs a forward pass of temporal spiking activity to update the eligibility traces. In a subsequent forward pass, intermediate layer features are projected onto auxiliary ANN modules. A decoupling module then integrates teacher signals, which are jointly optimized with the ground-truth labels to supervise the corresponding substructures.

pass is conducted based on temporal spike encoding without constructing the computational graph. During this phase, temporal information is utilized to accumulate the running statistics (mean and variance) for batch normalization (BN) layers, while the eligibility traces $e_{ij}^t$ of the spiking neurons are simultaneously updated. Definition of eligibility trace: $e_{ij}^t = \sum_{\tau=0}^{t} \lambda^{t-\tau} \cdot \frac{\partial S_i^t}{\partial V_i^\tau} \cdot \frac{\partial V_i^\tau}{\partial W_{ij}}$, $\lambda$ is the decay factor, $S_i^t$ denotes the spike output of neuron $i$ at time step $t$, and $V_i^\tau$ represents the membrane potential. The auxiliary branch remains inactive during this stage. In the second stage, a rate-based forward pass is performed, during which the eligibility traces computed in the first pass are employed to approximate the backward gradients: $\frac{\partial L}{\partial W_{ij}} \triangleq \frac{\partial L}{\partial r_i} \cdot \mathbb{E}[e_{ij}^t]$, where rate $r_i = \mathbb{E}[S_i^t]$. Rate-coding primarily captures rate-based spatial features, which are inherently aligned with the spatial inductive bias of artificial neural networks (ANNs). As such, the encoded data exhibits strong compatibility with ANN architectures, requiring no additional processing. The auxiliary branch is designed using depthwise separable convolutions and is supervised by an additional loss signal, allowing more precise gradients to be propagated from the ANN branches to intermediate layers of the main network, thereby promoting more effective convergence. Throughout this process, only a single-step computational graph and a small set of eligibility traces are retained, substantially reducing both memory and computational costs associated with backpropagation.

### 3.3 Limitations of Standard Self-Distillation

In standard self-distillation frameworks, the teacher label is typically derived from the prediction of the final layer, under the assumption that deeper layers, benefiting from the full network capacity, produce higher-quality predictions. This teacher output is then used to supervise intermediate layers. Formally, for a deep neural network with $L$ branches, let the output of each branches $M_l$ at training

---

**Algorithm 1** An algorithm with caption

---

**Require:** SNN model $f_{snn}$, timesteps T, hyper-parameter $\beta, \tau$ for rate-based backpropagation, Train datasets $D = \{x_i, y_i\}_{i=1}^n$, ANN block $A = \{A_1, A_2, ..., A_{L-1}\}$
**Ensure:** Train the SNN model based on self-distillation
 1: **for** each batch training data $D_i = \{x_i, y_i\}$ **do**     ▷ Spiking Forward
 2:     Update eligibility traces $e_{ij}^t$
 3: **end for**
 4: **for** each batch training data $D_i = \{x_i, y_i\}$ **do**     ▷ Rate Forward
 5:     Compute each sub-model outputs $\{p_1, p_2, ..., p_L\}$
 6:     Aggregate the teacher labels $y_{teacher}$ (8)
 7:     Compute the self-distillation loss as shown in Equation(10)
 8:     Compute the CE loss as shown in Equation (9)
 9:     Compute the final loss using Equation (11)
10:     Approximate backpropagation with eligibility traces $e_{ij}^t$.
11: **end for**

---

iteration $t$ be denoted as $\mathbf{p}_l^{(t)}$. The quality of the prediction at layer $l$ can be evaluated using the cross-entropy loss $\mathcal{L}_{\text{CE}}(\mathbf{p}_l^{(t)}, y)$ with respect to the ground truth label $y$.

Standard self-distillation assumes that the final layer $L$ consistently yields the best approximation of the ground truth across all layers:

$$\mathcal{L}_{\text{CE}}(\mathbf{p}_L^{(t)}, y) \leq \mathcal{L}_{\text{CE}}(\mathbf{p}_l^{(t)}, y), \quad \forall l \in \{1, \ldots, L-1\} \tag{3}$$

ensuring that the distillation process guides student layers toward a high-quality target distribution.

However, learning dynamics within deep networks are complex. Different layers may converge or reach peak performance at different training stages. In particular, shallow layers with fewer parameters may converge or saturate earlier than deeper layers. We define a binary indicator to denote whether the final layer provides the best prediction at iteration $t$:

$$\delta(t) = \begin{cases} 1, & \text{if } \mathcal{L}_{\text{CE}}(\mathbf{p}_L^{(t)}, y) \leq \min_{l \in \{1, ..., L-1\}} \mathcal{L}_{\text{CE}}(\mathbf{p}_l^{(t)}, y) \\ 0, & \text{otherwise} \end{cases} \tag{4}$$

The proportion of training iterations during which the final layer is not the best predictor is given by:

$$\eta = 1 - \frac{1}{T} \sum_{t=1}^{T} \delta(t) \tag{5}$$

In our preliminary experiments, we found that $\eta = 0.507$, indicating that in more than 50% of the training iterations, the final layer prediction is not globally optimal.

When suboptimal teacher predictions are used, they can lead to negative transfer for the student layers. Let $\mathbf{q}^{(t)}$ denote the teacher label at time $t$, and $\mathbf{p}_k^{(t)}$ denote the output of student branches $M_k$. The knowledge distillation loss is defined as:

$$\mathcal{L}_{\text{KD}}^{(t)} = \text{KL}\left(\mathbf{q}^{(t)} \parallel \mathbf{p}_k^{(t)}\right) \tag{6}$$

When a significant portion of teacher labels are inferior to the student predictions—as we observed empirically with $\eta = 0.507$—the expected distillation loss can be decomposed as:

$$\mathbb{E}_t\left[\mathcal{L}_{\text{KD}}^{(t)}\right] = (1-\eta) \cdot \mathbb{E}\left[\mathcal{L}_{\text{KD}}^{\text{good}}\right] + \eta \cdot \mathbb{E}\left[\mathcal{L}_{\text{KD}}^{\text{bad}}\right] \tag{7}$$

$\mathcal{L}_{\text{KD}}^{\text{good}}$ denotes the knowledge distillation loss under standard conditions. If $\mathcal{L}_{\text{CE}}(\mathbf{q}^{(t)}, y) > \mathcal{L}_{\text{CE}}(\mathbf{p}_k^{(t)}, y)$, then the teacher $\mathbf{q}^{(t)}$ is logically inferior to the student and should not be used as a supervisory signal, in this case, we define the loss as $\mathcal{L}_{\text{KD}}^{\text{bad}}$. Since the KL divergence term guides $\mathbf{p}_k^{(t)}$ toward $\mathbf{q}^{(t)}$, this misguidance may lead to performance degradation. Moreover, when the erroneous teacher prediction has high confidence in the incorrect class, its misleading impact is further amplified.

These observations call into question the effectiveness of always using the final layer output as a teacher and motivate the development of more robust teacher selection strategies during training.

### 3.4 Reliability-Separated Self-Distillation

In our training framework, auxiliary branches connected to intermediate layers produce multiple outputs, each of which can be regarded as an independent student in an online self-distillation setup. This naturally provides a rich set of candidate teacher signals. A straightforward solution is to generate the teacher label by computing a weighted average of the predictions from all n students. From the perspective of ensemble learning, increasing the number of voting models generally improves the stability of the final prediction. However, since the quality of teacher signals improves gradually during training, directly aggregating all predictions in the early training stages may result in unreliable teacher labels. To address this issue, we propose a reliability-separated self-distillation strategy. Specifically, we hypothesize that correctly predicted samples are more likely to exhibit reliable distributions compared to incorrectly predicted ones. As shown in Figure 2, we filter out samples that are incorrectly predicted by individual students and aggregate only the correctly predicted ones to form the final teacher label, following the ensemble learning principle. During loss computation, samples that are misclassified by all n students are excluded from the distillation process to ensure that only reliable teacher signals participate. While we do not deny that unreliable signals can still act as regularizers—preventing the student model from becoming overconfident in certain categories—we apply a small standard regularization term to these samples during loss calculation. This preserves the regularization effect of the original distillation process. Our strategy thus guides the student model to focus on learning from reliable knowledge while avoiding the misleading effects of noisy or unreliable signals.

$$y_{\text{teacher}} = \frac{\sum_{l=1}^{L} p_l \cdot \mathbb{I}\left(\arg\max p_l = \arg\max y\right)}{\sum_{l=1}^{L} \mathbb{I}\left(\arg\max p_l = \arg\max y\right) + \epsilon} \tag{8}$$

In the given formulation, $p_l$ denotes the predicted distribution of the $i$ student. The overall loss function of the model consists of two components: the hard loss $L_{ce}$ computed between each classifier's output $p_l$ and the ground truth label, and the soft distillation loss $L_{esd}$ derived from knowledge distillation. The definitions of the hard loss and the soft distillation loss are as follows:

$$L_{ce} = \sum_{l=1}^{L} \left[ -\sum_{c=1}^{C} y^{(c)} \log\left(p_l^{(c)}\right) \right] \tag{9}$$

$$L_{esd} = \sum_{l=1}^{L} \left\{ \left[ \sum_{c=1}^{C} p_{\text{teacher}}^{(c)} \log\left(\frac{p_{\text{teacher}}^{(c)}}{p_i^{(c)}}\right) \right] \cdot \mathbb{I}\left(\sum_{c=1}^{C} \left|p_{\text{teacher}}^{(c)}\right| \neq 0\right) + \eta \cdot \mathcal{R}_l \cdot \mathbb{I}\left(\sum_{c=1}^{C} \left|p_{\text{teacher}}^{(c)}\right| = 0\right) \right\} \tag{10}$$

In the aforementioned formula, $C$ represents the total number of categories, while $p_{teacher}^c$ denotes the teacher label's prediction value for the $c^{th}$ category of each sample. $\mathcal{R}_l$ represents the regularization applied to the remaining part, $\eta$ is used to control the weight of the regularization. Additionally, we introduce a balancing factor $\beta$, which is used to control the weight of the distillation loss.

$$L_{target} = L_{ce} + \beta \cdot L_{esd} \tag{11}$$

## 4 Experiments

### 4.1 Main Results

We compare our enhanced self-distillation framework with existing direct training methods across multiple classification benchmarks, including static datasets, CIFAR-10 [36], CIFAR-100 [36], and ImageNet [9] (Table 1), as well as neuromorphic datasets CIFAR10-DVS [40] (Table 2). We set $\beta = 0.3$ to control the weight of the auxiliary global distillation loss.

**Results on static datasets.** Experimental results demonstrate that our method achieves significant accuracy improvements over RateBP. On the CIFAR-100 dataset, it outperforms RateBP by 1.18% and 1.31% using ResNet-18 and ResNet-19, respectively, and achieves a 0.71% improvement on the ImageNet dataset. Our method approaches the performance of BPTT-based direct training and distillation approaches that rely on externally pre-trained artificial neural networks (ANNs). In contrast, our framework requires no external ANN teacher models; instead, it leverages high-quality knowledge generated by the model's own auxiliary branches to guide and enhance the training

Table 1: Comparison of top-1 accuracy (%) averaged over three runs on CIFAR-10, CIFAR-100, and ImageNet datasets. *indicates the use of an additional pre-trained ANN model for distillation. For all experiments on ImageNet, the ResNet-34 model is consistently used for training.

| Datasets | Training | Method | Architecture | Timestep | CIFAR10 Top-1 Acc (%) | CIFAR100 Top-1 Acc (%) | ImageNet Top-1 Acc (%) |
|---|---|---|---|---|---|---|---|
| Direct-training | OTTT [69] | online | VGG-11 | 6 | 93.52 | 71.05 | 65.15 |
| | OS [86] | online | ResNet-19 | 4 | 95.20 | 77.86 | 67.54 |
| | Dspike [42] | BPTT | ResNet-19 | 6 | 94.25 | 74.24 | 68.19 |
| | | | | 4 | 93.66 | 73.35 | |
| | | | | 2 | 93.13 | 71.68 | |
| | TET [11] | BPTT | ResNet-19 | 6 | 94.50 | 74.72 | 64.79 |
| | | | | 4 | 94.44 | 74.47 | |
| | | | | 2 | 94.16 | 72.87 | |
| | SEW-ResNet [17] | BPTT | ResNet-34 | 4 | - | - | 67.04 |
| | DSR [48] | one-step | PreAct-ResNet-18 | 20 | 95.10 | 78.50 | 67.74 |
| | RateBP [77] | one-step | ResNet-18 | 6 | 95.9 | 79.02 | 70.01 |
| | | | | 4 | 95.61 | 78.26 | |
| | | | | 2 | 94.75 | 75.97 | |
| | | | ResNet-19 | 6 | 96.38 | 80.83 | |
| | | | | 4 | 96.26 | 80.71 | |
| | | | | 2 | 96.23 | 79.87 | |
| w/ distillation | BKDSNN* [71] | BPTT | ResNet-19 | 4 | 94.64 | 74.95 | 67.21 |
| | SAKD [25] | BPTT | ResNet-19 | 4 | 96.06 | 80.10 | - |
| | TKS [15] | BPTT | ResNet-19 | 4 | 96.35 | 79.89 | 69.60 |
| | SM [12] | BPTT | ResNet-18 | 4 | 96.04 | 79.49 | 68.25 |
| | | | ResNet-19 | 4 | 96.82 | 81.70 | |
| | TWKD* [78] | BPTT | ResNet-18 | 6 | 95.96 | 79.80 | 71.04 |
| | | | | 4 | 95.57 | 79.10 | |
| | EAGD* [72] | one-step | ResNet-18 | 6 | 96.14 | 79.40 | 70.64 |
| | | | | 4 | 95.92 | 78.85 | |
| | | | | 2 | 95.19 | 77.06 | |
| | | | ResNet-19 | 2 | 96.56 | 81.44 | |
| | **ours** | one-step | ResNet-18 | 6 | $96.19 \pm 0.12$ | $80.20 \pm 0.17$ | 70.72 |
| | | | | 4 | $95.92 \pm 0.03$ | $79.30 \pm 0.21$ | |
| | | | | 2 | $95.29 \pm 0.10$ | $77.46 \pm 0.17$ | |
| | | | ResNet-19 | 4 | $96.39 \pm 0.01$ | $81.90 \pm 0.20$ | |
| | | | | 2 | $96.31 \pm 0.07$ | $80.97 \pm 0.05$ | |

process. Notably, these performance gains are achieved with constant memory consumption during backpropagation, resulting in a 75.80% reduction in memory usage and a 23.30% reduction in time consumption compared to BPTT training.

**Results on neuromorphic datasets.** On the CIFAR10-DVS neuromorphic dataset, BPTT-based methods and one-step training methods show differing performance characteristics. While BPTT preserves temporal dynamics more thoroughly, one-step methods focus more on spatiotemporal feature extraction. For example, methods such as [72, 77] may partially compress the temporal dimension. Consequently, BPTT tends to be more adaptable to neuron-based datasets. To address this challenge, we decouple the backpropagation component of the rate coding mechanism in our method and conduct a fair comparison using the same encoding scheme as prior work. Compared to RateBP, our method achieves a 1.50% performance gain under the same number of timesteps. Furthermore, when the rate coding component is decoupled, our method also achieves leading performance on the CIFAR10-DVS dataset, indicating its effectiveness under the temporal coding framework as well.

## 4.2 Ablation study

**Ablation study on self-distillation.** In this section, we analyze the effectiveness of the enhanced self-distillation method on spike-rate-coded SNNs and ANNs. We decouple the self-distillation component within the framework and examine the feedback from classifiers at different depths. Experimental results demonstrate that our self-distillation method offers significant advantages over standard self-distillation, and this observation also holds true for ANNs.

Table 2: Performance comparison of top-1 accuracy (%) on CIFAR10-DVS and ImageNet, averaged over three experimental runs.

| Training | Method | Architecture | Timestep | Top-1 ACC(%) |
|---|---|---|---|---|
| OTTT [69] | online | VGG-11 | 10 | 76.63 |
| RateBP [77] | one-step | ResNet-18 | 10 | 80.40 |
| EAGD [72] | one-step | ResNet-19 | 4 | 80.54 |
| TET [11] | BPTT | VGGSNN | 10 | 83.17 |
| SM [12] | BPTT | ResNet-18 | 10 | 83.19 |
| Enof [22] | BPTT | ResNet-19 | 10 | 80.10 |
| TWKD [78] | BPTT | ResNet-19 | 10 | 83.80 |
| **ours** | one-step | ResNet-18 | 10 | 81.40 |
| | | ResNet-19 | 10 | 81.90 |
| | BPTT | ResNet-18 | 10 | 85.70 |
| | | ResNet-19 | 10 | 85.90 |

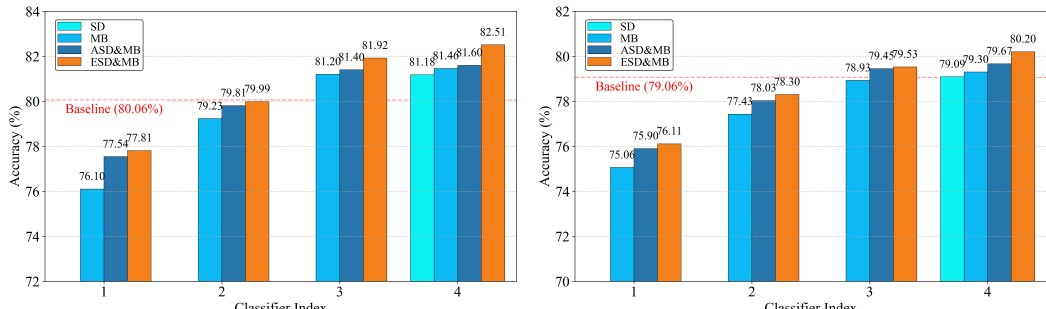

(a) Comparison of RateBP-Based Distillation Methods. (b) Comparison of ANN-Based Distillation Methods.

Figure 3: Ablation bar chart of the self-distillation module on ANNs and SNNs.

All distillation losses are computed based on KL divergence. We conduct comparative experiments on the CIFAR-100 dataset using a ResNet-18 model with a temporal dimension of T=6. The following methods are compared. SD: Introduces a pre-trained ANN teacher model with the same architecture and performs standard end-to-end distillation based on logits; MB: Adds lightweight auxiliary branches after each layer to facilitate the optimization of intermediate representations; ASD: Uses the final layer's predicted labels as teacher signals for intermediate layers; ESD: Decouples the reliability of student-generated labels and aggregates them for self-distillation training. As shown in Figure 3a and Figure 3b, all methods outperform the original baseline. Standard end-to-end distillation provides limited benefit to model training, highlighting the effectiveness of optimizing intermediate layers via auxiliary branches. The multi-branch self-distillation framework achieves superior performance without requiring any external ANN teacher models. Comparing ASD and ESD reveals that ESD avoids the misleading effects caused by unreliable predictions from the final layer in ASD, resulting in significant performance improvements. Our method improves the final classifier accuracy by 0.53%, and boosts the ANN model performance by 0.59%.

**Ablation of the Regularization Component.**
In general, knowledge distillation is considered to serve two main purposes. First, it guides the student's predictions toward a higher-quality distribution. Second, the soft labels act as a form of regularization, preventing the model from becoming overly confident in a single target class during training. When all n classifiers make incorrect predictions for a particular sample, it indicates

| Baseline | ESD(w/o Reg) | ESD |
|---|---|---|
| 79.02% | 80.20% | 80.32% |

Table 3: Ablation of the Regularization

Table 4: Comparison of average spike frequency across time steps on CIFAR100 using ResNet-18.

|  | Method | $T = 1$ | $T = 2$ | $T = 3$ | $T = 4$ | $T = 5$ | $T = 6$ | avg |
|---|---|---|---|---|---|---|---|---|
| Trained for 4 time steps | BPTT | 0.1799 | 0.2137 | 0.2045 | 0.2091 | - | - | 0.2018 |
|  | **ours** | 0.1591 | 0.1709 | 0.1715 | 0.1706 | - | - | 0.1680 |
| Trained for 6 time steps | BPTT | 0.1761 | 0.2034 | 0.2023 | 0.1966 | 0.2060 | 0.1941 | 0.1964 |
|  | **ours** | 0.1548 | 0.1560 | 0.1550 | 0.1516 | 0.1550 | 0.1532 | 0.1543 |

that the model has a systematic bias in feature representation at a global level. In such cases, the model is more likely to assign high confidence to incorrect classes. We acknowledge that even when the teacher's soft labels are unreliable, they can still play a role in smoothing the target distribution. Therefore, in our method, we impose a standard regularization term on this subset of samples. As shown in Table 3, the model achieves better generalization on the test set after applying regularization. It is worth noting, that in the middle to later stages of training, the teacher soft labels can cover the vast majority of samples, making the additional regularization relatively minor in proportion.

### 4.3 Performance on Energy Efficient Implementation

**Impact of time expansion.** We compare the proposed method with RateBP and BPTT in terms of the impact of varying time steps on training accuracy and the memory overhead during backpropagation. As shown in Figure 4b, the accuracy of our method continues to improve with an increasing number of time steps, clearly demonstrating its scalability in the temporal dimension. Since the auxiliary branch only participates in the second forward pass and still uses spike-rate encoding, the additional memory and computational overhead introduced during backpropagation remains constant. Figure 4a further illustrates that our method effectively decouples the memory cost of backpropagation from the time step, ensuring that training costs do not increase with larger values of T.

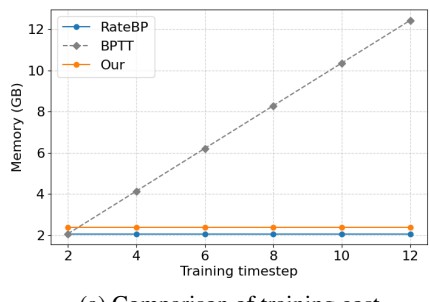

(a) Comparison of training cost.

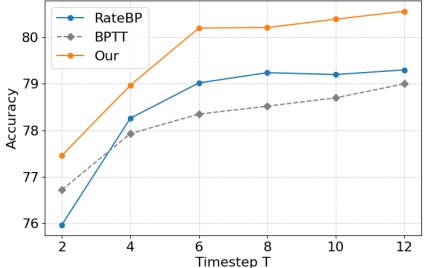

(b) Comparison of test performance.

Figure 4: Comparison of training cost and test performance of time steps.

**Spike firing rate analysis.** AS shown in the table 4, our method significantly reduces the spike frequency at each time step compared to the BPTT-based approach. This indicates that our method achieves sparser neural activity and lower energy consumption while maintaining competitive performance, aligning better with the original design goals of Spiking Neural Networks (SNNs).

## 5 Conclusion

This study proposes an enhanced self-distillation framework for efficiently training spiking neural networks (SNNs). Compared to the conventional BPTT approach, our method significantly reduces memory and time consumption during training, while leveraging auxiliary ANN branches to mitigate gradient errors in intermediate layers. Moreover, by selectively integrating high- and low-reliability predictions from multiple classifiers, we construct high-quality teacher signals that enable the model to better absorb valuable self-generated knowledge. This addresses a critical issue in standard self-distillation, where unreliable teacher labels can hinder the student model from converging in the

right direction. Experimental results demonstrate that our method achieves high-performance SNN training even under highly constrained computational settings.

## 6 Acknowledgments

This work was supported by the National Natural Science Foundation of China (Grant No. 62304203), international campus of ZJU international research collaboration seed project, and the ZJU-YST joint research center for fundamental science.

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

# Appendix

## A    Experimental Settings

### A.1    Datasets

#### A.1.1    CIFAR-10 and CIFAR-100

The CIFAR-10 and CIFAR-100 [36] datasets consist of color images with a resolution of 32×32 pixels and cover a range of object categories. Both datasets are released under the MIT license. CIFAR-10 contains 60,000 images across 10 classes, with 50,000 images for training and 10,000 for testing. CIFAR-100 includes 100 classes. Both datasets are preprocessed with zero-mean and unit-variance normalization. Data augmentation strategies follow AutoAugment [7] and Cutout[16], consistent with recent works [41, 5, 23, 66, 12]. At each time step, the raw pixel values are directly encoded [55] and fed into the network input layer.

#### A.1.2    ImageNet

The ImageNet-1K dataset [9] consists of 1,281,167 training images and 50,000 validation images across 1,000 distinct categories, and is available for non-commercial use. All images are standardized to have zero mean and unit variance. During training, images are randomly resized and cropped to 224×224 pixels, followed by horizontal flipping. For validation, images are first resized to 256×256 pixels and then center-cropped to 224×224. Similar to the treatment of CIFAR datasets, each image is converted into a temporal sequence via direct encoding before being fed into the network.

#### A.1.3    CIFAR10-DVS

The CIFAR10-DVS dataset [40] is a neuromorphic version of CIFAR-10, consisting of 10,000 event-based images captured by a Dynamic Vision Sensor (DVS) camera. The images have an increased spatial resolution of 128×128 pixels and are released under the CC BY 4.0 license. We split the dataset into 9,000 training images and 1,000 test images. Data preprocessing involves accumulating events into frames [19, 18] and downsampling the spatial resolution to 48×48 via interpolation. Additional data augmentation includes random horizontal flipping and random translation within a 5-pixel range, consistent with previous studies [49, 69].

### A.2    Training Setup

Our experiments adopt a sigmoid-based surrogate gradient method [19] to approximate the Heaviside step function, defined as $h(x, \alpha) = \frac{1}{1+e^{-\alpha x}}$, where the parameter $\alpha$ is set to 4. We follow the time-approximate backpropagation strategy from [77], and all implementations are based on the PyTorch [53] and SpikingJelly [19] frameworks. Experiments on CIFAR-10, CIFAR-100, CIFAR10-DVS, and ImageNet datasets are conducted on an NVIDIA GeForce RTX 3090 GPU. For all tasks, we use stochastic gradient descent (SGD) with a momentum of 0.9 [58], and apply a cosine annealing schedule [46] for learning rate adjustment. Additional hyperparameters are listed in the table 5.

Table 5: Hyperparameters Settings.

|  | CIFAR-10 | CIFAR-100 | ImageNet | CIFAR10-DVS |
|---|---|---|---|---|
| Epoch | 300 | 300 | 100 | 300 |
| Learning rate | 0.1 | 0.1 | 0.2 | 0.1 |
| Batch size | 128 | 128 | 512 | 32 |
| Weight decay | 5e-4 | 5e-4 | 2e-5 | 5e-4 |

### A.3    Network Architectures

For the CIFAR-10, CIFAR-100, and CIFAR10-DVS datasets, we adopt ResNet-18 and ResNet-19 as the backbone models. For the ImageNet dataset, ResNet-34 is used as the backbone model. All spiking neural network (SNN) models employ leaky integrate-and-fire (LIF) neurons, with the membrane potential decay factor uniformly set to 0.5. The implementation is based on the activation-driven paradigm proposed by [17].

Table 6: Performance under different $\beta$ using ResNet-18 on CIFAR100.

| classifier/$\beta$ | 0.0 | 0.1 | 0.3 | 0.5 | 0.7 | 0.9 | 1.0 |
|---|---|---|---|---|---|---|---|
| 1 | 75.06 | 75.71 | 76.11 | 75.91 | 76.12 | 76.20 | 76.50 |
| 2 | 77.74 | 76.96 | 78.30 | 77.70 | 77.51 | 77.81 | 77.60 |
| 3 | 78.93 | 79.29 | 79.53 | 79.64 | 79.89 | 79.57 | 79.10 |
| 4 | 79.30 | 79.69 | 80.20 | 79.96 | 79.89 | 79.61 | 79.47 |

### A.4 Surograte Branches Design

In the design of the auxiliary branches, we need to balance computational complexity and feature extraction capability. On one hand, if the auxiliary branch is too simple, it may fail to effectively extract features from the intermediate layers of the model, thereby affecting training performance. On the other hand, if the auxiliary branch is too complex, it will significantly increase computational complexity, which contradicts the low training cost characteristic of rate coding and may also lead to gradient fragmentation issues.

We first consider standard convolution, whose parameter storage complexity and computational complexity are given by:

$$P_{std} = D_k^2 \times C_{in} \times C_{out} \tag{12}$$

$$C_{std} = D_k^2 \times C_{in} \times C_{out} \times D_f^2 \tag{13}$$

$D_k$ represents the kernel size, $C_{in}$ and $C_{out}$ denote the number of input and output channels, respectively, and $D_f$ is the size of the output feature map. Compared to traditional convolution, depthwise separable convolution decouples computation along the spatial and channel dimensions. Depthwise convolution extracts spatial features, followed by pointwise convolution, which captures channel-wise features. Its parameter storage complexity and computational complexity are given by:

$$P_{dsc} = D_k^2 \times C_{in} + C_{in} \times C_{out} \tag{14}$$

$$C_{dsc} = D_k^2 \times C_{in} \times D_f^2 + C_{in} \times C_{out} \times D_f^2 \tag{15}$$

The ratio of the number of parameters required for depthwise separable convolution to that of standard convolution is $\frac{1}{N} + \frac{1}{D_k^2}$, and the ratio of computational complexity is also $\frac{1}{N} + \frac{1}{D_k^2}$. This demonstrates that depthwise separable convolution requires fewer parameters and has faster computation, aligning with the low training cost characteristic of rate coding.

Additionally, to prevent issues such as gradient explosion and enhance training performance, we introduce residual connections in the auxiliary branches. Finally, after passing through the classifier, we obtain the soft labels of the required submodule.

## B   More Results

### B.1   Selection of Parameter $\beta$

In Table 6, We fixed the weight of the hard loss to 1.0 and focused on investigating the impact of varying the self-distillation weight $\beta$ on model training by evaluating the ResNet-18 model on the CIFAR-100 dataset. We observe that as $\beta$ increases from 0, the overall model performance improves progressively, with accuracy steadily rising across all classifiers. This indicates that larger values of $\beta$ enable each classifier to better learn from the teacher's knowledge, thereby facilitating the training of the entire network. The model achieves optimal performance when $\beta = 0.3$. However, when $\beta$ exceeds 0.3, we observe that while some intermediate classifiers continue to improve in accuracy, the final classifier's performance begins to decline. This suggests that an excessively large distillation weight shifts the training focus more toward the intermediate layers, diverging from the optimal learning direction of the final classifier. As a result, this imbalance leads to a gradient diversion problem, hindering the overall performance of the model.

Table 7: Time and memory overhead at different timesteps.

| Method | Eval Metrics | T=2 | T=4 | T=6 | T=8 | T=10 | T=12 |
|--------|--------------|-----|-----|-----|-----|------|------|
| BPTT | Time cost(s/batch) | 0.076 | 0.161 | 0.253 | 0.360 | 0.474 | 0.606 |
| | Memory(MB) | 2645.94 | 4889.32 | 7133.21 | 9383.85 | 11636.11 | 13887.80 |
| RateBP | Time cost(s/batch) | 0.106 | 0.143 | 0.184 | 0.227 | 0.267 | 0.311 |
| | Memory(MB) | 2120.71 | 2120.78 | 2121.18 | 2296.48 | 2747.20 | 3197.42 |
| Our | Time cost(s/batch) | 0.125 | 0.154 | 0.194 | 0.237 | 0.277 | 0.318 |
| | Memory(MB) | 2380.09 | 2380.76 | 2381.91 | 2383.06 | 2758.69 | 3208.44 |

## B.2 Standard self-distillation suboptimal solution count

We evaluate the classification performance of different depth classifiers on the CIFAR-100 dataset using ResNet-18. Instances where the deepest classifier performs worse than shallower classifiers are considered as invalid teacher label examples (as indicated by the boxed region in Figure 5). We conducted three random observation experiments, and the average proportion of invalid teacher labels was 50.7%. As analyzed in Section 3.3, these invalid teacher labels hinder the student model from converging in the correct direction. More notably, as the confidence in the target class of the mispredicted teacher labels increases, and the overall proportion of invalid labels rises during training, the negative impact intensifies. This phenomenon highlights the theoretical foundation for our proposed enhanced self-distillation method.

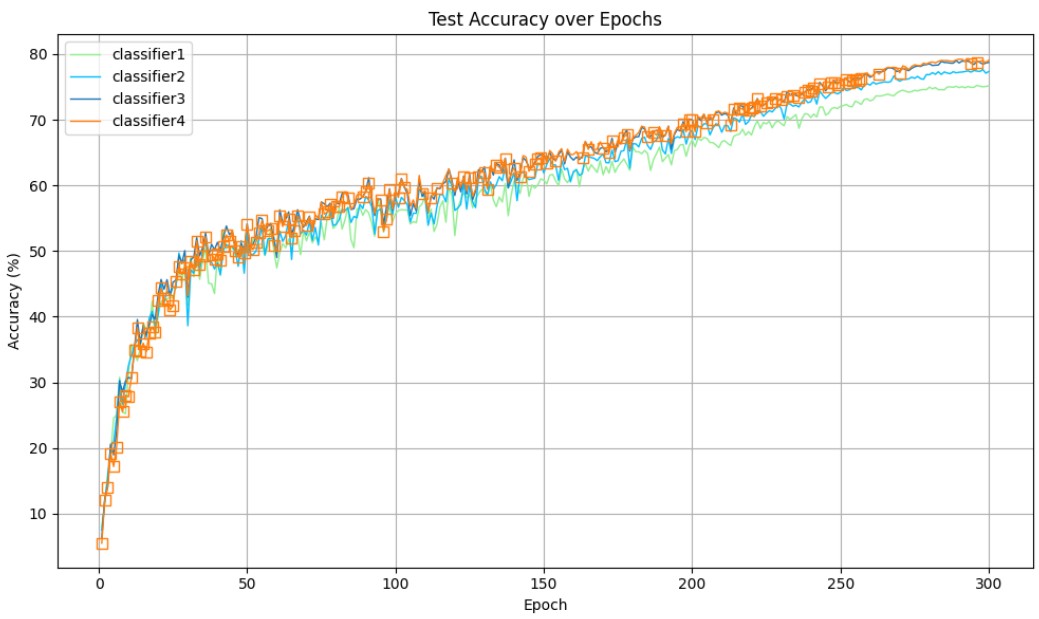

Figure 5: Classification performance of classifiers with different depths during the training process.

## B.3 Comprehensive Evaluation of Training Costs

As a supplement to Figures 1 and 4a in the main text, we provide the memory and time overhead at different timesteps for BPTT, RateBP, and our proposed method, as shown in Table 7.

## B.4 Impact on inference efficiency

We recognize SEENN[43] as a highly meaningful work. It determines whether to stop the inference process early by computing the confidence level over the first t time steps. This allows it to achieve performance close to that of the full T time steps within a shorter inference duration, thereby reducing

inference cost. As shown in Figures 6a and 6b, we further demonstrate that models trained under our framework are compatible with this method. We tested confidence thresholds of 0.7, 0.8, 0.9, 0.99, and 0.999. On CIFAR-10 and CIFAR-100, our method achieves the performance of 6 time steps using only an average of 2.4214 and 3.5371 time steps, respectively. Compared with RateBP, the accuracy improves by 0.48% and 1.18%, with only a slight increase of 0.0048 and 0.1438 time steps, respectively.

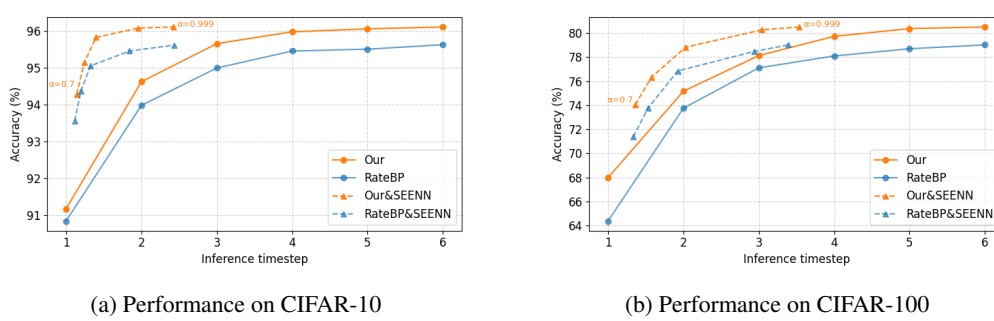

(a) Performance on CIFAR-10          (b) Performance on CIFAR-100

Figure 6: Comparison of SEENN performance

## B.5 Visualization Analysis

We provide a visual analysis comparing standard self-distillation with the enhanced self-distillation approach. We utilize t-SNE visualization to observe classifiers at different depths and compare the clustering effects between Standard Self-Distillation and our proposed method. As shown in Figure 7 and Figure 8, two key observations can be made: First, it is evident that the deeper the classifier, the more compact the clustering becomes. Second, classifiers trained with Enhanced Self-Distillation exhibit stronger class separability compared to those of the same depth trained with Standard Self-Distillation. These results confirm that each classifier benefits from learning high-quality teacher signals, effectively pushing the upper bound of performance.

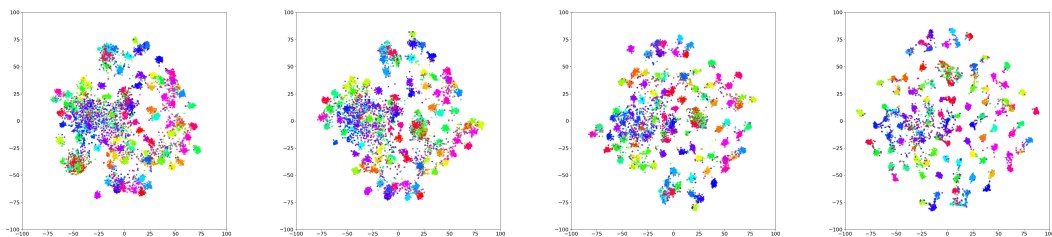

Figure 7: Clustering Patterns of Classifiers 1–4 in the Standard Self-Distillation

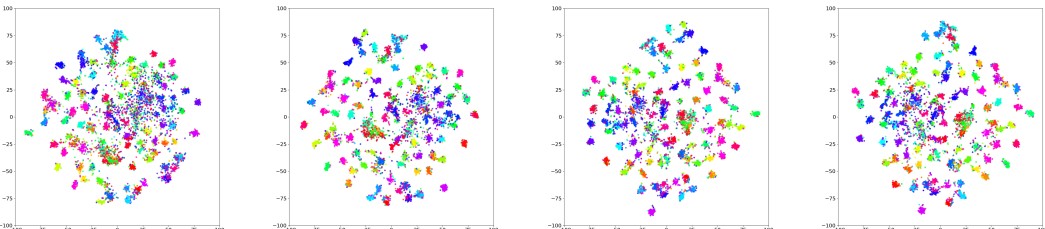

Figure 8: Clustering Patterns of Classifiers 1–4 in the Enhanced Self-Distillation

## B.6 Analysis of Rate Statistics

Our method is trained based on rate coding and incorporates an ANN branch to enhance training effectiveness. While this hybrid strategy improves model performance, it may also influence the

spiking characteristics inherent to traditional SNNs. To investigate this, we tracked the average spike frequency across different layers over multiple time steps. As shown in the figure 9, the results indicate that the average firing rate remains highly consistent over time, confirming the stability of the model during inference.

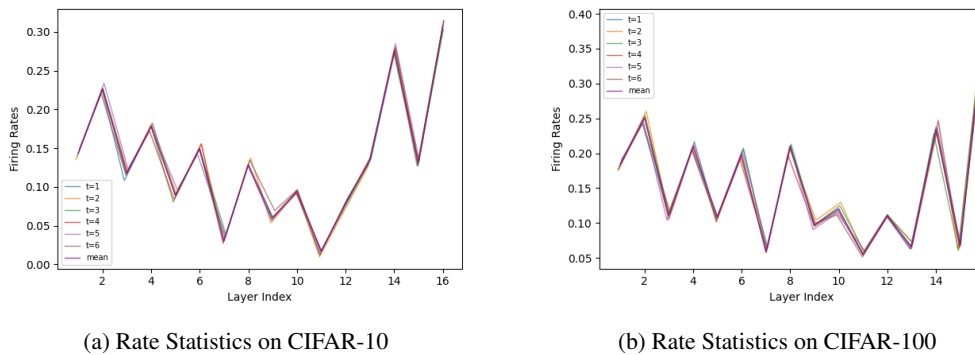

(a) Rate Statistics on CIFAR-10

(b) Rate Statistics on CIFAR-100

Figure 9: Comparison of SEENN performance

## C  Social Impacts and Limitations

This research primarily focuses on achieving high-performance training of Spiking Neural Networks (SNNs) under limited training conditions, and therefore does not lead to direct negative social impacts. Compared to Artificial Neural Networks (ANNs), SNNs inherently offer lower energy consumption during inference, which helps reduce carbon dioxide emissions. The proposed method uses a rate coding-based training approach, which contributes to reducing the training time and hardware requirements for SNNs.

The paper acknowledges several challenges regarding the proposed method. First, by introducing a lightweight artificial neural network branch based on rate encoding, the training cost is slightly higher than that of RateBP. However, these memory costs are fixed constants and do not increase linearly with the time steps, as is the case with backpropagation through time (BPTT). More importantly, the performance breakthroughs achieved through experimental evaluations make this additional cost worthwhile. Furthermore, frequency-based backpropagation is designed to efficiently capture spatiotemporal feature representations to optimize training, but its performance on sequence tasks is weaker than that of BPTT. This is discussed in Section 4.1. To ensure a fair comparison, we decoupled the frequency-based backpropagation training method on a dataset with neuron morphology. Experimental results show that our improved self-distillation method demonstrates outstanding performance on both frequency- and time-based backpropagation methods, as well as on artificial neural networks.

