# OpenReview forum: "Enhanced Self-Distillation Framework for Efficient Spiking Neural Network Training"
_NeurIPS.cc/2025/Conference — NeurIPS 2025 poster_

### Official Review · Reviewer_mbYz · 2025-06-24

**Clarity:** 3
**Significance:** 4
**Originality:** 3
**Rating:** 5
**Confidence:** 5

**Summary:**

This paper presents an enhanced self-distillation framework for efficient training of Spiking Neural Networks (SNNs). The key ideas include projecting the firing rates of intermediate SNN layers onto lightweight ANN branches and using a reliability-separated self-distillation strategy to ensure that only high-quality knowledge guides the optimization process.

**Questions:**

1.The paper proposes an enhanced self-distillation framework but does not analyze its convergence in detail. It is recommended that the authors theoretically analyze the convergence rate and convergence conditions of the proposed method, especially the effect on convergence when dealing with unreliable teacher signals.
2.The influence of different β values on the model performance is shown in Table 6 in the text. It indicates that this method is relatively sensitive to the selection of β values and there exists an optimal range of β values. However, the specific method on how to select the optimal β value is not provided in the text.
3.The author should consider whether a dynamic reliability assessment mechanism can be introduced, such as how to adjust the reliability assessment criteria of teacher signals as the training progresses.
4.In Section 3.2, the author mentions the use of eligibility traces to approximate gradients in backpropagation. However, the accuracy and potential errors of this approximate method are not analyzed in detail in the text.

**Ethical Concerns:**

["NO or VERY MINOR ethics concerns only"]

**Final Justification:**

Thank you very much for your reply. I still maintain the original rating.

**Limitations:**

Please reference Weaknesses and Questions.

**Paper Formatting Concerns:**

No.

**Quality:**

3

**Strengths And Weaknesses:**

Strength:
1.The proposed enhanced self-distillation framework and the idea of using rate-based backpropagation combined with a reliability-separated self-distillation strategy are innovative and addresses a critical issue in traditional self-distillation methods where unreliable teacher labels can hinder convergence.
2.The authors provide extensive experimental results on multiple datasets, demonstrating the effectiveness of their method. The improvements in accuracy, memory consumption, and training time compared to BPTT and other methods are substantial and well-documented.
3.This paper presents a detailed theoretical analysis of the effects of reliable and unreliable predictions of teacher signals on student models.
Weaknesses:
1.The reliability separation strategy proposed in the article may ignore some samples that have potential value in the early stage of training but are currently misclassified in practical operation. For example, in Table 6, when the β value increases from 0.3 to 0.5, although the performance of the classifier in the middle layer improves, the performance of the final classifier begins to decline. This indicates that simply filtering out misclassified samples may cause the model to miss some important learning signals.
2.It is mentioned in the text that the auxiliary branch uses depth-separable convolution to reduce the computational complexity. However, although this design has advantages in computational efficiency, it may be insufficient in feature extraction ability.
3.The text mentions the formation of teacher labels by filtering out samples with correct predictions (see Section 3.4), but does not elaborate on how to define the criteria for "correct predictions." For example, is it based on the classification accuracy rate or the confidence threshold?

---

> ### Author Rebuttal · Authors · 2025-07-30
>
> ## 1.Potential Value of Error Classification Samples (to weakness1)
> Thank you for your suggestions. Please refer to the response to Reviewer 4txP entitled "6.Analysis of the Reliability of Error Labels (to w4)---The quality of incorrect labels  ".
> ## 2.Introduction of a Dynamic Reliability Assessment Mechanism (to weakness1, question3)
> Thank you for your suggestions. Please refer to the response to Reviewer 4txP entitled "6.Analysis of the Reliability of Error Labels (to w4)---A scheme for differentiating reliability based on confidence ".
> ## 3.Why choose depthwise separable convolution(to w2)
> We compared depthwise separable convolution (DWConv) and standard convolution (StdConv). While StdConv is theoretically better at feature extraction, DWConv achieved 0.53% higher accuracy in our experiments. This supports our design, as the auxiliary module aims to optimize substructures. As discussed in Supplementary A.4, StdConv may overly focus on the auxiliary branch, shifting optimization away from the final classifier. This is why we use DWConv in auxiliary branches.
> ||FC1|FC2|FC3|FC4|
> |-|-|-|-|-|
> |DWConv|76.11|78.3|79.53 |80.2|
> |StdConv|76.42| 77.91| 79.62 |79.67|
>
> In addition, Supplementary Section A.4 analyzes the theoretical advantages of DWConv over StdConv in terms of compute and memory efficiency, supporting our goal of training high-accuracy models with limited resources.
> ## 4.Criteria for "Correct Prediction" (to w3)
> The criterion for "correct prediction" is classification accuracy, if the class with the highest predicted probability among the n classes matches the target label, the prediction is considered correct and thus reliable.
> ## 5.Impact of Unreliable Teacher Signals on Convergence (to question1)
> Thank you for your suggestion. We first define: the standard convex objective $L(\theta) = E_{(x,y)}[\ell(f_\theta(x), y)],$ where $\ell$ is the cross-entropy loss and $y$ is the ground-truth label. Its correct gradient direction is $g_{\text{true}}(\theta) = \nabla_\theta\\ell(f_{\theta_t}(x), y),$ and the parameter update is: $\theta_{t+1} = \theta_t - \eta\ g_{\text{true}}(\theta), \quad \theta_t \to \theta^* \quad \text{as } t\to\infty.$ In distillation, by introducing a soft label to assist training, the student model learns richer inter-class distribution information; however, when this soft label is unreliable, it often backfires. We illustrate this with an extreme example: When the teacher label is incorrect and the confidence in the wrong class is too high ($p_{\text{err}}\to1$), the erroneous gradient can be approximated as: $g_{\text{err}}(\theta) = \nabla_\theta \ell(f_\theta(x), \hat{y}\_{\text{wrong}}).$ When this error becomes too large, it will cause the model to deviate from its original optimization direction; this is precisely why we need to separate the reliable parts of the labels. Thank you again for your suggestion. We will include a more detailed derivation in future work.
> ## 6.Optimal Range of β Values（to q2）
> Thank you for your question. To determine the behavior of β, we conducted fine-grained validation experiments on CIFAR-10 and CIFAR-100. Both datasets showed similar patterns. As noted in Supplementary Section B.1, increasing β from 0 enhances the effect of self-distillation, but beyond a certain limit, the accuracy of the main classifier declines.This is likely because an excessively large loss weight shifts the training focus toward intermediate classifiers, neglecting the optimization of the main classifier. The optimal β values for CIFAR-10 and CIFAR-100 are around 0.5 and 0.3, respectively. Therefore, we believe setting β within the range [0.3, 0.5] can effectively improve training stability.
> |β|0|0.1|0.3|0.5|0.7|0.9|1.0|
> |-|-|-|-|-|-|-|-|
> |CIFAR10|95.88|96.06|96.09|96.21|96.04|95.97|95.95|
> |CIFAR100|79.30|79.69|80.20|79.96|79.89|79.61|79.47|
> ## 7.Error analysis of rate approximation (to q4)
> Thank you for your question. Here, we provide a concise overview of how rate coding achieves gradient approximation. A detailed exposition will be included in the supplementary materials of the forthcoming updated version. First, we define discrete time steps as $t\in\{1,\dots,T\}$ and layer index as $l$. Let membrane potential be $u^l_t$, spike output $s^l_t$, threshold $V_{th}$, decay $\lambda$, the weight matrix of layer $l$ as $W^l$, and the Heaviside step function as $H(\cdot)$. The network output at time $t$ is $o_t = W^L s^L_t$, the final prediction $y_{\mathrm{pred}} = \frac{1}{T}\sum_{t=1}^T o_t$, and the loss function is defined as $\mathcal{L} = \ell\bigl(y_{\mathrm{pred}},\,y\bigr).$ The spike generation process:  $u^l_t = \lambda\bigl(u^l_{t-1} - V_{th}\,s^l_{t-1}\bigr)+ W^l\,s^{\,l-1}\_{t-1},s^l_t = H\bigl(u^l_t - V_{th}\bigr).$
> **BPTT gradient computation for membrane potential:**
> $$
> \frac{\partial \mathcal{L}}{\partial \mathbf{u}^l_t}  = \frac{\partial \mathcal{L}}{\partial \mathbf{s}^l_t} \frac{\partial \mathbf{s}^l_t}{\partial \mathbf{u}^l_t} + \sum_{\tau > t} \frac{\partial \mathcal{L}}{\partial \mathbf{s}^l_{\tau}} \frac{\partial \mathbf{s}^l_{\tau}}{\partial \mathbf{u}^l_{\tau}} \prod_{i=\tau-1}^t \left( \frac{\partial \mathbf{u}^l_{i+1}}{\partial \mathbf{u}^l_i} + \frac{\partial \mathbf{u}^l_{i+1}}{\partial \mathbf{s}^l_i} \frac{\partial \mathbf{s}^l_i}{\partial \mathbf{u}^l_i} \right)
> $$
> Weight gradient accumulation: $\nabla_{W^l}\mathcal{L}
> = \sum_{t=1}^T \frac{\partial \mathcal{L}}{\partial u^l_t}s^{l-1}\_t$, and gradient propagation to the previous layer:$\frac{\partial \mathcal{L}}{\partial s^{\_l-1}\_t}=\frac{\partial \mathcal{L}}{\partial u^l_t}\_W^{l\top}.$ Under the rate-coding assumption, information in an SNN can be represented by the average firing rate. The average firing rate per layer (denoted $r^l$) is computed as the expectation of spikes over time:  $r^l = \mathbb{E}[s^l_t] \approx \frac{1}{T} \sum_{t=1}^T s^l_t.$ In the linear propagation stage, each time-step input $I^l_t = W^l s^{l-1}\_{t-1}$ is approximated by its average $c^l = \mathbb{E}[I^l_t] = W^l r^{l-1}$. The gradient for the linear part is straightforward:$\frac{\partial r^{l-1}}{\partial c^{l}} = {W^{l}}^\top.$ Next, we need to define the relationship between the average input $c^l$ and the firing rate $r^l$, $\frac{\partial r^{l}}{\partial c^{l}}$. Starting from the exact gradient, the influence of different time-step inputs on the output is expressed as:
> $$
> \frac{\partial \mathbf{s}\_\tau^l}{\partial \mathbf{I}\_t^l} = \frac{\partial \mathbf{s}\_\tau^l}{\partial \mathbf{u}\_t^l} =
> \begin{cases}
> \frac{\partial \mathbf{s}\_\tau^l}{\partial \mathbf{u}\_\tau^l} \prod_{i=\tau-1}^t \left( \frac{\partial \mathbf{u}\_{i+1}^l}{\partial \mathbf{u}\_i^l} + \frac{\partial \mathbf{u}\_{i+1}^l}{\partial \mathbf{s}\_i^l} \frac{\partial \mathbf{s}\_i^l}{\partial \mathbf{u}\_i^l} \right) & \text{if } \tau \ge t \\
> \frac{\partial \mathbf{s}\_t^l}{\partial \mathbf{u}\_t^l} & \text{if } \tau = t \\
> 0 & \text{if } \tau < t
> \end{cases}
> $$
> By accumulating this dynamic process over time, we obtain the influence of a single time-step input on the overall spike output:
> $$
> \mathbf{\kappa}^l_t = \sum_\tau \frac{\partial \mathbf{s}\_\tau^l}{\partial \mathbf{I}\_t^l} = \left( \frac{\partial \mathbf{s}\_t^l}{\partial \mathbf{u}\_t^l} + \sum_{\tau > t} \frac{\partial \mathbf{s}\_\tau^l}{\partial \mathbf{u}\_\tau^l} \prod_{i=\tau-1}^t \left( \frac{\partial \mathbf{u}\_{i+1}^l}{\partial \mathbf{u}\_i^l} + \frac{\partial \mathbf{u}\_{i+1}^l}{\partial \mathbf{s}\_i^l} \frac{\partial \mathbf{s}\_i^l}{\partial \mathbf{u}\_i^l} \right) \right)
> $$
> Under rate coding, using the approximation $\frac{1}{T} \sum_{t=1}^{T} I^l_t \approx c^l$, we derive the surrogate gradient of neural dynamics via mean estimation:
> $$
> \left( \frac{\partial \mathbf{r}^l}{\partial \mathbf{c}^l}\right)\_{\text{rate}} \equiv \sum_\tau \left( \frac{\partial(\mathbb{E}[\mathbf{s}^l_t])}{\partial \mathbf{I}^l_\tau} \frac{\partial \mathbf{I}^l_\tau}{\partial \mathbf{c}^l} \right) =\frac{1}{T} \sum_t \sum_\tau \left( \frac{\partial \mathbf{s}^l_t}{\partial \mathbf{I}^l_\tau} \right) =\mathbb{E}[\mathbf{\kappa}^l_t]
> $$
> Using these compressed gradients, rate-based backpropagation can be performed by iterating only over spatial dimensions:
> $$
> \left(\frac{\partial \mathcal{L}}{\partial c^l}\right)\_\text{rate}
> = \left(\frac{\partial \mathcal{L}}{\partial c^L} \prod_{i=L-1}^{l} \left(\frac{\partial c^{i+1}}{\partial r^i} \left(\frac{\partial r^i}{\partial c^i}\right)\_\text{rate}\right)\right)
> = \left(\frac{\partial \mathcal{L}}{\partial c^L} \prod_{i=L-1}^{l} \left(W^{i^\top} \mathbb{E}\left[\kappa^l_t\right]\right)\right)
> $$
> In summary, the main difference between rate-based backpropagation and BPTT arises from the assumptions on the mean estimator representation of firing rates. By assuming temporal components are independent, the rate-coding approach establishes equivalence with BPTT. Denote $\delta_t(s^l) = \frac{\partial \mathcal{L}}{\partial s^l_t}$ as the gradient of spike $s^l_t$ computed via BPTT, and $\kappa_t^l = \sum_{\tau} \frac{\partial s_t^l}{\partial I_{\tau}^l}$ as the accumulated gradient of input $I_t^l$ on all spikes. If all layers satisfy $\mathbb{E}\left[\delta_t(s^l) \, \kappa_t^l\right] = \mathbb{E}\left[\delta_t(s^l)\right] \mathbb{E}\left[\kappa_t^l\right],$
> the gradient and accumulated response are statistically independent, then $\mathbb{E}\left[\delta_t(s^l)\right] = \left( \frac{\partial \mathcal{L}}{\partial \mathbf{r}^l} \right)\_{\text{rate}}.$  Further, let $\delta_t(I^l) = \frac{\partial \mathcal{L}}{\partial I_t^l}$ and assume $\mathbb{E}\left[\delta_t(I^l)\,s_t^{l-1}\right] = \mathbb{E}\left[\delta_t(I^l)\right]\mathbb{E}\left[s_t^{l-1}\right].$ Then the rate-based weight gradient equals the BPTT:$\left( \nabla_{W^l} \mathcal{L} \right)\_{\text{rate}} = \frac{1}{T} \nabla_{W^l} \mathcal{L}.$ Thank you again for your suggestion. A more detailed process will be updated in future versions.

---

### Official Review · Reviewer_4txP · 2025-06-26

**Clarity:** 2
**Significance:** 3
**Originality:** 2
**Rating:** 5
**Confidence:** 4

**Summary:**

The paper proposes a self-distillation framework based on the RateBP training framework, introducing a lightweight ANN auxiliary branch in the intermediate layers of the SNN to provide more accurate gradients for the intermediate layers, thereby alleviating the gradient distortion problem. In addition, the authors propose the ESD method, which determines the reliability of all branches’ output signals, and combines them into a final high-quality teacher signal for guiding the learning of the auxiliary branch. This paper achieves significant performance improvements and reduced training costs on all conducted datasets.

**Questions:**

1.The method in the paper is based on the soft reset LIF model. Can it be applied to the hard reset rules of LIF?

2.In Figure 3 (b), according to the paper, self-distillation alleviates the gradient error problem. Why are methods like MB, ASD, and ESD still effective on ANNs where there is no gradient error?

**Ethical Concerns:**

["NO or VERY MINOR ethics concerns only"]

**Final Justification:**

The authors provide a detailed rebuttal that completely addressed all my concerns. In the rebuttal, the authors found effective theoretical derivations for their method, enhancing the completeness of the paper. Meanwhile, I am convinced by their dual theoretical explanations for the effectiveness of the method on ANN and the brand-new ablation experiments, which successfully transformed a core concern into a highlight of the paper.

**Limitations:**

Yes

**Quality:**

2

**Strengths And Weaknesses:**

Strength

1.The paper achieves a simultaneous increase in training efficiency and network performance, representing an excellent attempt to solve the training problem of SNN.

2.This paper identifies potential problem with standard self-distillation in SNN and proposes a simple and effective solution,

Weakness

1.The paper lacks a detailed discussion of the auxiliary branches, such as the structure of the auxiliary branches, the insertion location in the main branch, and the relationship between various choices of auxiliary branches and the performance improvements.

2.Figure 2 is missing the caption. The current caption is a placeholder paragraph.

3.This paper need to provide an in-depth theoretical analysis (or experiment) of how gradient errors are corrected by auxiliary branches.

4.The definition of reliability may be overly simple. Sometimes, misclassification output can contain more useful information than correct classification output. The paper does not further explore more refined reliability measurement methods, which may still have significant room for improvement.

5.The contribution 1 of the paper seems to derive from RateBP method, and a comparative analysis with it should be added.

6.In Table 7, the BPTT memory cost at T=12 is excessively large. This may be caused by a transcription error.

---

> ### Author Rebuttal · Authors · 2025-07-30
>
> ## 1.Whether it is applicable to the hard reset of LIF. (to question1)
> Thank you for your question. The proposed method is applicable to the hard-reset LIF. In the code section of the supplementary materials, within the model.layer module, we define the function calcu_elig_factor_hard_reset to update the eligibility trace using the hard reset method. We have evaluated the performance of the baseline, RateBP, MB, and ESD methods separately. The detailed experimental results are as follows:
> |CIFAR100,T=6,ResNet18|baseline|RateBP|MB|ESD|
> |-|-|-|-|-|
> |Acc|78.41|78.34|79.38|79.76|
>
> ## 2.Why is our method still effective on artificial thermal networks. (to q2)
> This issue stems from the fact that our method can be understood from two theoretical perspectives.
>
> First, from the perspective of surrogate learning, a lossless structure (ANN in our case) provides more accurate gradients to the intermediate layers of the backbone, mitigating gradient errors. This aligns with the motivation of our method. Since ANN introduces no temporal approximation errors, the i-th auxiliary branch can be viewed as a surrogate module that supplies more accurate gradients to the preceding sub-network.
>
> Second, from the perspective of deep supervision: inserting additional supervision signals at intermediate layers and jointly optimizing them with the main loss improves gradient flow and model generalization. This also explains why our method is effective even when both the backbone and auxiliary branches are implemented with lossless ANN structures.
>
> Our approach incorporates both the surrogate learning perspective and the deep supervision perspective, which can be validated through a simple experiment. If we replace the ANN in the auxiliary module with an SNN that matches the backbone, keeping all other settings the same, and ignore gradient approximation errors, we refer to this result as SNN_Rate and observe that accuracy still improves by 0.4% and 0.44% over the baseline and RateBP, respectively. However, it is still lower than our method. This demonstrates that deep supervision is also effective, but our method enables more comprehensive optimization under rate-coded training.
> |baseline|RateBP|SNN_Rate|ESD|
> |-|-|-|-|
> |79.06|79.02|79.46|80.20|
>
> ## 3.Detailed analysis of the auxiliary branch. (to weakness1)
> Experimental setup: T=6, ResNet18, CIFAR100.
> ### Structural analysis
> As stated in Section 3.2 Model Training of the main paper, the auxiliary branch in our framework is constructed using depthwise separable convolutions. The specific architecture is as follows:DepthwiseConv (stride=2, Cin → Cin) → PointwiseConv (Cin → Cin) → BN → ReLU → DepthwiseConv (stride=1, Cin → Cin) → PointwiseConv (Cin → Cout) → BN → ReLU.
> ### Why choose depthwise separable convolution
> We compared depthwise separable convolution (DWConv) and standard convolution (StdConv). While StdConv is theoretically better at feature extraction, DWConv achieved 0.53% higher accuracy in our experiments. This supports our design, as the auxiliary module aims to optimize substructures. As discussed in Supplementary A.4, StdConv may overly focus on the auxiliary branch, shifting optimization away from the final classifier. This is why we use DWConv in auxiliary branches.
> ||FC1|FC2|FC3|FC4|
> |-|-|-|-|-|
> |DWConv|76.11|78.3|79.53 |80.2|
> |StdConv|76.42| 77.91| 79.62 |79.67|
>
> In addition, Supplementary Section A.4 analyzes the theoretical advantages of DWConv over StdConv in terms of compute and memory efficiency, supporting our goal of training high-accuracy models with limited resources.
> ### Regarding the position of the auxiliary branch
> In the paper, to facilitate evaluating the effectiveness of our framework, we uniformly set three auxiliary branches inserted after the ResNet convolutional blocks [2, 4, 6].
>
> In practice, the insertion positions can be treated as hyperparameters, allowing for increased design flexibility and better scalability when applied to more complex models. We also experimented with alternative insertion schemes, such as placing branches after block [4] alone, or after blocks [3, 6]. Fewer insertion points benefit training efficiency but may limit model accuracy. More flexible auxiliary branch insertion options will be provided in our forthcoming open-source code.
> ||[4]|[3,6]|[2,4,6]|
> |-|-|-|-|
> |Acc(%)|79.65|79.92|80.2|
> |Time(s)|0.195|0.199|0.208|
> |Memory(MB)|2401.7|2430.10|2601.0|
>
> ## 4.Figure 2 lacks a title.（to w2）
> Thank you for your careful review. We will include this clarification in the next version.
> ## 5.How the auxiliary branch mitigates gradient errors. (to w3)
> We take one auxiliary branch as an example and denote its output as $y_{i}$, with the final target output as $y_{c}$. From experiments, we observe that KL$(y_{i}|y_{c})$ eventually converges, and $y_{c}$ approaches $y_{i}$. We set $y_{1}=y_{c}+\xi$, where $\xi$ is a small error term. Define $\frac{\partial y_{1}}{\partial \theta } = S$, $\frac{\partial y_{c}}{\partial \theta } = G$, where $G$ is the expected gradient of the backbone network.
>
> Due to rate approximation and surrogate gradient errors, we define $R\left(\frac{\partial y_{c}}{\partial \theta }\right) = G+\delta$, so the error ratio is $K_{rate} = \frac{\delta}{G}$. We define the total loss as $L_{total} = \alpha \cdot L_{1}+L_{c}$, where $\frac{\partial L_{1}}{\partial y_{1}}= \frac{\partial L_{c}}{\partial y_{c}}+{\xi }'$. It can be derived that the ideal total gradient of our method is
>
> $$
> \frac{\partial L_{total}}{\partial \theta}= (1+\alpha )\cdot \frac{\partial L_{c}}{\partial y_{c}}\cdot G+\alpha \cdot \frac{\partial L_{c}}{\partial y_{c}}\cdot (S-G)+\alpha \cdot {\xi }'\cdot S,
> $$
>
> where ${\xi }'$ is a higher-order infinitesimal of $G$ and can be neglected, thus
>
> $$
> \frac{\partial L_{total}}{\partial \theta}= \frac{\partial L_{c}}{\partial y_{c}}\cdot (G+\alpha \cdot S).
> $$
>
> Therefore, under our framework, the error ratio
>
> $$
> K_{esd}=\frac{\delta }{G+\alpha \cdot S  } \le \frac{\delta}{G} =K_{rate}.
> $$
>
> Therefore, under our framework, the error rate $K_{esd}$ is always less than $K_{rate}$.
> Through experiments, we found that setting $\alpha$=1.0 yields the best performance. An excessively large loss weight for the auxiliary branches may hinder the convergence of the backbone network.
> |α| 0.0 | 0.5 |0.7|0.9|1.0|1.2|1.5
> |-|-|-|-|-|-|-|-|
> |Acc(%)|78.96|79.17|79.21|79.24|79.30|79.28|78.89|
>
> ## 6.Analysis of the Reliability of Error Labels. (to w4)
> ### The quality of incorrect labels
> We acknowledge that incorrect predictions may carry useful information. However, in self-distillation, especially early in training, correct predictions are generally closer to high-quality label distributions. We show this by comparing their KL divergence to a reliable distribution.
>
> We introduced a ResNet34 model with an accuracy of up to 82%, which we consider to have high-quality label A. We define the correctly predicted portion of the output as B1 and the incorrectly predicted portion as B2. Across many training epochs, it is observed that KL(A, B1) is consistently smaller than KL(A, B2), especially in the early training phase. This strongly indicates that the correctly predicted part is more likely to be close to a high-quality label distribution than the incorrectly predicted part.
> |epoch|10|20|50|100|150|200|300|
> |-|-|-|-|-|-|-|-|
> |KL(A,B1)|0.2119|0.1821|0.1745|0.1678|0.1405|0.1396|0.1306|
> |KL(A,B2)|0.3117|0.297|0.2628|0.2676|0.296|0.237|0.1756|
> ### A scheme for differentiating reliability based on confidence
> Additionally, based on your suggestion, we further refined the separation of incorrect teacher labels using confidence scores. Specifically, we introduced a threshold k, if an incorrect prediction has a confidence score for the wrong class below k, it is reclassified as a reliable label. This refinement is an extension of the analysis in Section 3.3, which indicates that incorrect labels with high confidence in the wrong class are more misleading to the student. We treated k as a hyperparameter, and experimental results show that when k<0.03, the accuracy remains above 80%. However, a noticeable drop occurs when k>0.03.
> |k|0|0.02|0.03|0.04|0.05|1.0|Average Confidence|
> |-|-|-|-|-|-|-|-|
> |Acc|80.2|80.04|80.12|79.87|79.89|79.71|80.21|
>
> We take into account that label confidence dynamically changes during training. As shown in the experiments, the average confidence increases progressively over the course of training.
> |epoch|10|50|100|150|200|250|
> |-|-|-|-|-|-|-|
> |Average Confidence|0.0355|0.0466|0.0494|0.0525|0.0556|0.0598|80.21|
>
> A fixed k cannot adapt to the changing confidence levels at different training stages. Therefore, we introduced a dynamic thresholding mechanism based on average confidence setting k=Average Confidence. This dynamic approach shows greater potential compared to a fixed threshold, but in practice, the improvement over our original results is limited.This demonstrates that our method, while simple in design, is highly effective in practice. In summary, while we do not deny that better reliability separation strategies may exist, the key contribution of our work is introducing the concept of reliability in the context of self-distillation. Defining more accurate reliability criteria remains an important direction for future research.
> ## 7.Comparison with RateBP. (to w5)
> We show performance results in Tables 1–2 and Figures 3–4(b), and training cost in Figure 1, Figure 4(a), and Table 7. Our method outperforms RateBP while keeping most of its cost benefits.
> ## 8.Abnormal memory record at T=12. (to w6)
> Thank you for your careful review. We found that the abnormal memory usage under the BPTT setting with T=12 was caused by redundant intermediate variables. We have re-evaluated the peak memory consumption during training using torch.cuda.max_memory_allocated, and the updated average values are shown in the table below.
> |T|2|4|6|8|10|12|
> |-|-|-|-|-|-|-|
> |Memory(MB)|2645.9|4889.3|7133.2|9383.8|11636.1|13887.8|

---

> > ### Comment · Reviewer_4txP · 2025-08-02
> >
> > Thank you for providing such a detailed, rigorous, and insightful rebuttal. The explanation from "surrogate module learning" and "deep supervision" perspectives, supported by a clever ablation study to disentangle these effects, is highly convincing. At the same time, I believe that the concise mathematical derivation of gradient error can provide theoretical support for your method.

---

### Official Review · Reviewer_e4GK · 2025-07-02

**Clarity:** 2
**Significance:** 2
**Originality:** 2
**Rating:** 3
**Confidence:** 5

**Summary:**

This paper proposes a novel training framework for Spiking Neural Networks (SNNs) that enhances performance and efficiency by integrating self-distillation with rate-based backpropagation. The key idea is to project firing rates from intermediate SNN layers into lightweight ANN branches, enabling substructure optimization via self-generated knowledge. To address the issue of unreliable knowledge in standard self-distillation, the authors introduce a decoupling mechanism that filters out low-quality guidance, ensuring only reliable signals influence training.

**Questions:**

1. Could the authors clarify whether the proposed method is able to preserve any spike timing information during the knowledge transfer process, or is all temporal structure reduced to rate-based statistics?
2. Could the proposed distillation framework remain effective if the auxiliary ANN branches were replaced with SNN-based substructures, thereby maintaining a fully spike-based architecture?

**Ethical Concerns:**

["NO or VERY MINOR ethics concerns only"]

**Limitations:**

Yes.

**Paper Formatting Concerns:**

None.

**Quality:**

3

**Strengths And Weaknesses:**

Strengths:
1. Improved Training Efficiency: The proposed approach shows noticeable reductions in memory usage and training time on standard benchmarks, which may be beneficial for resource-constrained SNN training scenarios.
2. Refined Self-Distillation Strategy: The idea of decoupling reliable and unreliable self-generated knowledge offers a thoughtful refinement to conventional self-distillation methods, potentially improving training stability.
3. Good Empirical Coverage: The method is evaluated on multiple datasets, including both static and neuromorphic benchmarks, and shows some promise for generalization, including preliminary applicability to ANN models.

Weaknesses:
1. Limited Exploitation of Temporal Dynamics: The proposed method does not fully leverage the intrinsic temporal modeling capabilities of SNNs. Instead, it primarily relies on ANN-based distillation mechanisms and rate-based approximations to enhance performance, which may overlook the rich temporal information encoded in precise spike timing that is critical for neuromorphic tasks.
2. Inconsistency Between SNN and ANN Pathways: Projecting SNN firing rates into ANN branches introduces a modality mismatch, as the ANN surrogate may not faithfully preserve the temporal characteristics of spiking activity.

---

> ### Author Rebuttal · Authors · 2025-07-30
>
> ## 1.Applicability of the distillation framework to other model structures. (to question 2)
> We appreciate your suggestion, we replaced the auxiliary ANN module with an SNN module and evaluated the effectiveness of our self-distillation framework under both rate-based and temporally-based training schemes. The corresponding experimental results are denoted as SNN_RateBP and SNN_BPTT, respectively.
> | T=6,CIFAR100|baseline|SNN_RateBP|SNN_BPTT|Our|
> | -------- | -------- | -------- |-------- |-------- |
> |Acc(%)| 79.06| 79.46 |80.14|80.20|
> |Memory(MB)|7132.2|2906.1|7676.3|2381.91|
> |Time(s)|0.253|0.273|0.331|0.208|
>
> From the experimental results on CIFAR100, using rate-based training (SNN_RateBP) achieves a 0.4% accuracy improvement over the baseline. However, due to the replacement of the ANN module, the approximation error introduced by rate coding is no longer properly compensated, leading to a 0.74% drop in accuracy compared to our original method. Furthermore, the auxiliary branch must now follow the same information integration mechanism as the backbone SNN, making it less lightweight than the ANN variant. As a result, both efficiency and computational cost are inferior to our original design.
>
> The method also proves effective under BPTT-based training (SNN_BPTT), where no approximation error from rate coding exists. This version achieves a 1.08% improvement over the baseline, showing performance comparable to our proposed method. Additionally, as reported in Table 2 of the main paper, it also outperforms previous approaches on neuromorphic datasets. However, this approach loses the low-cost advantage of rate coding, resulting in a linear increase in memory consumption with respect to the temporal dimension, which limits its scalability under constrained computational resources.
>
> In summary, our framework achieves efficient training by utilizing temporally averaged information, focusing on spatially effective features. While we acknowledge that this may compromise the ability to model fine-grained temporal dynamics, the use of rate-based training decouples the linear dependency between the temporal dimension and memory consumption during backpropagation. This offers a reliable approach to training high-accuracy models efficiently under resource-constrained scenarios. Moreover, our proposed self-distillation framework is generalizable to other architectures, including ANNs and temporally trained SNNs.
>
>
>
> ## 2.Whether any spike timing information is retained during the knowledge transfer process.（to question1）
> Thank you for your question. As described in Section 3.2 of the paper, in our framework, the first forward pass is conducted based on temporal dynamics, during which the full temporal statistics are preserved and the eligibility traces are updated. In the second forward pass, both the backbone network and the auxiliary branch operate entirely based on rate information, effectively simplifying the process to rate-based statistical computation.
>
> However, as discussed in Question 2, the first and second contributions of the paper are independent. The self-distillation framework remains applicable to training methods based on BPTT.
> ## 3.Limited utilization of temporal dynamics (to weakness1)
> Thank you for your insight. Our rate-based framework achieves efficient training by integrating the mean of temporal information, with a focus on spatially effective features. As discussed in the supplementary material Section C Social Impacts and Limitations, we acknowledge that this approach compromises the model's ability to capture fine-grained temporal dynamics, which is an important area for improvement in future work.
>
> However, the rate-based training paradigm decouples the linear relationship between the temporal dimension and memory consumption during backpropagation. This provides a highly cost-effective solution for training high-accuracy models under resource-constrained scenarios.
>
> Moreover, as mentioned in our response to Question 2 and further validated by additional experiments, our self-distillation framework is also effective in temporally trained SNNs and ANNs, further demonstrating the general applicability of our approach.
> ## 4.Inconsistencies between SNN and ANN pathways (to weakness2)
> Thank you for your question. In the rate coding framework, we aggregate the mean of temporal information into rate-based representations to enable efficient training. During the first forward pass, which is based on temporal dynamics, only the statistics and eligibility traces required by the backbone network are collected. The ANN module is not involved at this stage, so there is no need to address the mapping between SNN and ANN.
>
> In the second forward pass, since the backbone network operates on rate-based representations—and rate coding is highly compatible with ANN modules, both emphasizing spatial information—there is no issue of feature mismatch between the two.
>
> It is important to clarify that the rate-based forward pass focuses solely on spatial-domain information, which means that the auxiliary ANN branch does not need to preserve any spike-based temporal features.

---

> > ### Comment · Reviewer_e4GK · 2025-08-07
> >
> > Thank you for your response. While some concerns were clarified, the hybrid SNN-ANN approach still deviates from the core principle of spike-based information transmission in SNNs. I therefore maintain my original rating.

---

> > > ### Author Response · Authors · 2025-08-08
> > >
> > > Thank you for your suggestion. We believe this issue can be discussed from two perspectives.
> > >
> > > First, we fully acknowledge the importance of spike-based information transmission on neuronal datasets. However, in most static scenarios, temporal dynamics are not strictly required. Our experimental results support this view: our method (ESD) outperforms contemporary approaches in terms of accuracy, including those that strictly follow temporal-dynamic principles [1,2], while incurring significantly lower computational overhead. Therefore, we believe our method offers a viable solution for training high-accuracy models under resource constraints, particularly in static-task settings where temporal modeling is not essential.
> > >
> > > ||SM[1]|TWSD[2]|RateBP[3]|EAGD[4]|ESD|
> > > | -| -| -|- |-| - |
> > > |Acc(%)|79.49|79.80|79.02|79.40|82.20|
> > > |Memory(MB)|9228.00|9105.71|2120.78|2504.10|2381.91|
> > > |Time(s)|26.7|26.9|18.4|21.1|19.4|
> > >
> > > Moreover, when facing dynamic tasks such as neuronal datasets, our self-distillation framework can be effectively extended to models that leverage spike-based information transmission. As shown in our experimental results, our method still outperforms a range of contemporary approaches on DVS-CIFAR10 [1,2,5,6]. While this may introduce linear overhead, it does not undermine the fact that our self-distillation framework demonstrates strong performance even when applied to temporally trained spiking models.
> > >
> > > ||TET[5]|SM[1]|TWKD[2]|ENOF[6]|ESD|
> > > | -| -| -|- |-| - |
> > > |Acc(%)|83.17|83.19|83.80|80.50|85.70|
> > >
> > > In summary, our method can be viewed in two versions. For standard static scenarios, the rate-based self-distillation framework provides a low-cost solution for training high-accuracy models. In contrast, for dynamic neuronal tasks, decoupling the rate components enables the model to exploit the advantages of spike-based information transmission. Compared to contemporary approaches, our method consistently achieves superior accuracy in both settings. Each version demonstrates clear advantages under different task conditions, and unifying these strengths represents an important direction for future research.
> > >
> > > Second, from the perspective of ANN-to-SNN conversion, this line of work aims to avoid the non-differentiability of spikes in direct training by mapping a pre-trained artificial neural network (ANN) into a spiking neural network (SNN) [7,8,9,10,11,12,13]. This is a well-established and widely recognized research paradigm. Since the original ANN is inherently static and lacks temporal structure, the resulting SNN—while spike-based in form—typically transmits information in a manner that still closely approximates rate coding. Although this approach does not explicitly encode temporal dynamics through precise spike timing, it remains a significant and influential direction in SNN research due to its training efficiency and scalability.
> > >
> > > Our method essentially integrates the strengths of both direct training and conversion-based approaches. It retains the cost-efficiency of conversion during training, while also incorporating the low inference latency and high accuracy commonly associated with directly trained SNNs.
> > >
> > > Furthermore, we would like to clarify that our method relies on rate-based approximation only during the training phase; during inference, it adopts a fully spike-based design.
> > >
> > > Thank you once again for your valuable suggestion.
> > >
> > > [1] Surrogate Module Learning: Reduce the Gradient Error Accumulation in Training Spiking Neural Networks.
> > >
> > > [2] Efficient Logit-based Knowledge Distillation of Deep Spiking Neural Networks for Full-Range Timestep Deployment.
> > >
> > > [3] Advancing Training Efficiency of Deep Spiking Neural Networks through Rate-based Backpropagation.
> > >
> > > [4] Efficient ANN-Guided Distillation: Aligning Rate-based Features of Spiking Neural Networks through Hybrid Block-wise Replacement.
> > >
> > > [5] Temporal Efficient Training of Spiking Neural Network via Gradient Re-weighting.
> > >
> > > [6] EnOF-SNN: Training Accurate Spiking Neural Networks via Enhancing the Output Feature.
> > >
> > > [7] A New ANN-SNN Conversion Method with High Accuracy, Low Latency and Good Robustness.
> > >
> > > [8] TCL: an ANN-to-SNN Conversion with Trainable Clipping Layers.
> > >
> > > [9] Optimal ANN-SNN Conversion for Fast and Accurate Inference in Deep Spiking Neural Networks.
> > >
> > > [10] Optimal ANN-SNN Conversion for High-accuracy and Ultra-low-latency Spiking Neural Networks.
> > >
> > > [11] Faster and Stronger: When ANN-SNN Conversion Meets Parallel Spiking Calculation.
> > >
> > > [12] An all integer-based spiking neural network with dynamic threshold adaptation.
> > >
> > > [13] A Unified Optimization Framework of ANN-SNN Conversion: Towards Optimal Mapping from Activation Values to Firing Rates.

---

> ### Author Response · Authors · 2025-08-06
>
> Dear Reviewers,
>
> We sincerely appreciate your time and effort in reviewing our manuscript and providing valuable feedback.
>
> As the author-reviewer discussion phase draws to a close, we would like to confirm whether our previous responses have adequately addressed your concerns. We submitted detailed replies a few days ago and hope they have resolved the issues you raised.
>
> If you require any further clarification or have additional questions, please do not hesitate to contact us. We remain fully open to continued discussion.
>
> Best regards,

---

### Official Review · Reviewer_zqGh · 2025-07-03

**Clarity:** 2
**Significance:** 3
**Originality:** 2
**Rating:** 4
**Confidence:** 3

**Summary:**

The paper proposes an enhanced self-distillation framework for efficient Spiking Neural Network training, addressing the high memory and time costs of Backpropagation Through Time. The experiments on CIFAR-10, CIFAR-100, CIFAR10-DVS, and ImageNet demonstrate that the method reduces training complexity while achieving high-performance SNN training.

**Questions:**

In addition to the questions above, there are also the following points：

1. Unclear notation. Some symbols lack proper definitions and explanations. Such as the meanings of $L_{KD}^{good}$ and $L_{KD}^{bad}$ in Eq(7) are missing.

2. Formatting Issues: The highest performance values in the tables should be highlighted in bold for better readability. Table 3 should include percentage signs (%) for clarity.

3. Can you give more details about how to decouple the teacher signal into reliable and unreliable components in your methods?

**Ethical Concerns:**

["NO or VERY MINOR ethics concerns only"]

**Final Justification:**

4 Borderline accept. The authors give clear explanations of whether their method belongs to self-distillation. And they add many comparisions experiments with existing works. They solved my previous questions.

**Limitations:**

yes

**Paper Formatting Concerns:**

No issues.

**Quality:**

3

**Strengths And Weaknesses:**

Strength:
1. The paper proposed a novel method for efficient spiking neural network training.

2. This method has good empirical results. It outperforms previous methods on static/neuromorphic datasets

Weakness:
1. Is this truly self-distillation? Since it involves SNN-to-ANN conversion before distillation.

2. The method's effectiveness remains unverified for modern spiking neural architectures like Spiking Transformers.

3.Insufficient literature review on self-distillation. The paper missed some key references and experiment comparisons with previous self-distillation methods, weakening the contextual grounding of the proposed approach.

---

> ### Author Rebuttal · Authors · 2025-07-30
>
> ## 1.Whether our framework belongs to self-distillation. (to weakness1)
>
> Thank you for your question. It is reasonable to consider whether this architecture conforms to the definition of self-distillation from the perspective of the conversion between SNN and ANN. Admittedly, our network does incorporate a transition from SNN to ANN. However, in our model, the auxiliary ANN module is integrated as a substructure within the overall architecture, which distinguishes our approach from conventional distillation methods [1][2] that typically rely on an additional, separately pre-trained teacher model. In contrast, our approach generates teacher labels directly from an internal substructure of the model itself. On this basis, we contend that our architecture is consistent with the concept of self-distillation. Furthermore, prior works such as [3][4] have adopted similar strategies, using auxiliary branches to produce teacher labels, and these methods are widely recognized in the field as instances of self-distillation. Notably, in [4], the authors also implement a transition from a backbone SNN to an ANN branch during training, leveraging the outputs of the ANN component to guide the optimization of the backbone network. Their paper explicitly describes this approach as a form of self-distillation, which further supports our rationale.
>
> In addition, one of the contributions of our paper is addressing the issue in self-distillation frameworks where synchronous training of teacher and student signals leads to the teacher signal interfering with student learning in the early stages of training. We propose a solution that explicitly distinguishes between different levels of label reliability. This approach not only addresses the specific challenges identified in our study, but also has the potential to be generalized and applied to other self-distillation frameworks. This method enables a secondary performance breakthrough without introducing additional computational overhead, which further supports our claim that the proposed framework should be categorized as a self-distillation method.
>
> ## 2.The effectiveness of our method on modern spiking neural architectures such as Spiking Transformers. (to weakness2)
> Thank you for your valuable suggestion. We use Spikingformer as the baseline to reproduce three methods: RateBP [5], MB, and ESD. Through ablation studies, we demonstrate that our method remains effective on modern spiking neural architectures such as Spiking Transformers and exhibits generalizable patterns. Specifically, on the CIFAR100 dataset, with the time step set to T=4, we evaluate several configurations including Spikingformer, Spikingformer + RateBP, MB, and ESD. Here, MB introduces the same auxiliary branch as ESD but removes the self-distillation module. As shown in the experimental results, our method (ESD) achieves accuracy improvements of 0.92% and 0.81% over the baseline on the Spikingformer-4-384 and Spikingformer-4-256 models respectively, while reducing memory usage by 20.91% and 28.85%. Moreover, this advantage becomes more prominent as the number of time steps increases.
>
> In addition, the rate encoding error leads to a decrease in accuracy compared to the baseline, but the auxiliary branch helps mitigate this degradation. This observation aligns with the pattern described in Section 4.2 “Ablation study on self-distillation” of the main paper., RateBP < MB < ESD, which indicates that our method demonstrates consistent generalizability across modern spiking neural network architectures such as Spiking Transformers.
> |          | Spikingformer | RateBP  | MB      | ESD    |
> | :-------: | :------------: | :------: | :------: | :------: |
> | 4-384    |     79.09      |  77.29   |  79.54   |  80.01   |
> | 4-256    |     77.43      |  77.29   |  77.74   |  78.24   |
>
> ## 3.Citations of other self-distillation related work (to weakness3)
> Thank you for your valuable suggestion. We conducted a comparative study against several state-of-the-art self-distillation methods in the SNN domain in recent years. The evaluation was performed on the CIFAR100 dataset. Models marked with * use ResNet19, while the others are based on ResNet18. According to the experimental results, our framework achieves state-of-the-art performance among self-distillation methods in the SNN field. In addition, unlike other works, our rate-based training approach significantly reduces training costs, making it feasible to train models with higher temporal resolutions under limited computational resources.
> |SAKD*[7]|SM[8]|SM*|TSSD(T=2)[9]|TKS*[10]|Our|Our*|
> | -------- | -------- | -------- |-------- |-------- | -------- |-------- |
> |80.10|79.49| 81.70| 73.40 |79.89|80.20|82.14|
>
> ## 4.Regarding symbols and formatting (to question 1, 2)
> We appreciate your feedback on the notation and tables in our paper. You are correct that some symbols, such as those in Equation (7), were not clearly defined. We will revise the manuscript to clarify these definitions and improve the overall readability.
>
> As discussed in Section 3.3, in traditional self-distillation frameworks, the teacher and student labels are fixed. We evaluate their quality through the cross-entropy loss with the ground-truth label: $L_{\text{CE}}(q^{(t)}, y)$ and $L_{\text{CE}}(p^{(t)}, y)$. However, due to possible differences in convergence speed, the quality of the teacher label may become lower than that of the student during training: $L_{\text{CE}}(q^{(t)}, y) > L_{\text{CE}}(p^{(t)}, y)$. In this case, computing $\text{KL}(q^{(t)} \| p^{(t)})$ may cause the student to converge in the wrong direction. We define the KL divergence in this case as $L_{\text{KD}}^{\text{bad}}$. Conversely, if the teacher label is of higher quality than the student, we define it as $L_{\text{KD}}^{\text{good}}$.
>
> We will revise the equations and tables you pointed out in the next version. Thank you again for your detailed review — your suggestions help make this work more rigorous and complete.
>
>
> ## 5.More details about the decoupling component. (to question3)
> As mentioned in Section 3.4 of the paper, we assume that samples correctly predicted by the network are more likely to exhibit reliable distributional characteristics compared to those that are incorrectly predicted. Specifically, suppose the network has n classification heads: A₁, A₂, A₃...Aᵢ...Aₙ. For each classification head i, we divide the predicted batch into two categories: (1) samples for which the target class k is correctly predicted (class k has the highest predicted probability among all C classes), and (2) samples where the prediction for class k is incorrect.
>
> We then set the soft labels of the second category (incorrect predictions) entirely to zero, while keeping the soft labels of the correctly predicted samples unchanged, thus generating a new soft label set. To mitigate the issue where even correctly predicted samples may still exhibit unreliable distributions over non-target classes, we apply an averaging ensemble over the outputs of all n classification heads. This is based on ensemble learning theory [6], which suggests that increasing the number of voting models typically enhances the stability of the final prediction. In this process, only correctly predicted samples contribute to the voting; incorrectly predicted samples, having been zeroed out, contribute nothing to the final teacher label.The resulting teacher label, Teacher, is the ensemble of the n reliable predictions. In cases where all n classification heads mispredict a particular sample, the corresponding Teacher label for that sample will be all zeros. We interpret such instances as the model having likely fallen into a learning failure or ambiguity for those samples. Therefore, during the computation of the distillation loss, we use a masking mechanism to exclude these samples from distillation. However, we acknowledge that even unreliable labels can play a regularization role in the distillation process. To preserve this benefit, we apply a small regularization term to these excluded samples, preventing the model from becoming overconfident in learning from such ambiguous cases.
>
> The implementation details can be found in the supplementary materials, specifically in the layer.py script within the model module (lines 171–191), where we provide the full process of decoupling reliable and unreliable components and computing the final loss.
>
>
>
> ## Reference
> [1] Efficient Logit-based Knowledge Distillation of Deep Spiking Neural Networks for Full-Range Timestep Deployment
>
> [2] BKDSNN: Enhancing the Performance of Learning-based Spiking Neural Networks Training with Blurred Knowledge Distillation
>
> [3] Be Your Own Teacher: Improve the Performance of Convolutional Neural Networks via Self Distillation
>
> [4] Surrogate Module Learning: Reduce the Gradient Error Accumulation in Training Spiking Neural Networks
>
> [5] Advancing Training Efficiency of Deep Spiking Neural Networks through Rate-based Backpropagation
>
> [6] Online Knowledge Distillation via Collaborative Learning
>
> [7] SAKD: Sparse attention knowledge distillation
>
> [8] Surrogate Module Learning: Reduce the Gradient Error Accumulation in Training Spiking Neural Networks
>
> [9] Self-Distillation Learning Based on Temporal-Spatial Consistency for Spiking Neural Networks
>
> [10] Temporal Knowledge Sharing Enable Spiking Neural Network Learning From Past and Future

---

> ### Author Response · Authors · 2025-08-05
> **Supplementary Information Regarding the Rebuttal.**
>
> ## 2.The effectiveness of our method on modern spiking neural architectures such as Spiking Transformers. (to weakness2)
> The definitions of MB and ESD mentioned here are consistent with those in Section 4.2 (Ablation Study) of the main text.
> MB: Adds lightweight auxiliary branches after each layer to facilitate the optimization of intermediate representations.
> ESD: Decouples the reliability of student-generated labels and aggregates them for self-distillation training.
>
> ## 3.Citations of other self-distillation related work (to weakness3)
>
> In addition to the self-distillation methods discussed in the SNN domain, we also conducted a comparison with self-distillation approaches in the ANN domain. The results are as follows.
>
> |FRSKD[11]|BYOT[3]|USKD[12]|PS-KD[13]|CS-KD[14]|Our|
> | -------- | -------- | -------- |-------- |-------- | -------- |
> |80.49|78.64|79.90|79.24|78.72|82.51|
>
> Please feel free to let us know if there are any additional suggestions.
>
> [11] Refine Myself by Teaching Myself : Feature Refinement via Self-Knowledge Distillation
> [12] From Knowledge Distillation to Self-Knowledge Distillation: A Unified Approach with Normalized Loss and Customized Soft Labels
> [13] Self-Knowledge Distillation with Progressive Refinement of Targets
> [14] Self-distillation with Batch Knowledge Ensembling Improves ImageNet Classification
>
> ## 5.More details about the decoupling component. (to question3)
>
> In addition to the above discussions, we conducted a detailed experimental analysis on why incorrectly predicted samples are defined as unreliable labels.
> Please refer to the response to Reviewer 4txP entitled "6. Analysis of the Reliability of Error Labels (to w4) — The quality of incorrect labels."
>
> Furthermore, considering that incorrectly predicted samples may still carry useful knowledge, we further explored a more fine-grained scheme for separating reliability.
> Please refer to the response to Reviewer 4txP entitled "6. Analysis of the Reliability of Error Labels (to w4) — A scheme for differentiating reliability based on confidence."
>
> For a detailed discussion on the auxiliary branches,
> please refer to the response to Reviewer 4txP entitled "6. Analysis of the Reliability of Error Labels (to w4) — The quality of incorrect labels."
>
> Finally, we sincerely thank you once again for your valuable feedback on our work. We look forward to hearing from you and welcome any further discussions at any time.

---

> > ### Comment · Reviewer_zqGh · 2025-08-06
> >
> > Thank you for your responce. The authors have resolved my questions. I'd like to rise my score.

---

### Comment · Area_Chair_o4xp · 2025-08-06

Dear reviewers,

As reviewers, you are expected to stay engaged in discussion.

-  It is not OK to stay quiet.
-  It is not OK to leave discussions till the last moment.
-  If authors have resolved your (rebuttal) questions, do tell them so.
-  If authors have not resolved your (rebuttal) questions, do tell them so too.

Please note that, to facilitate discussions, Author-Reviewer discussions were extended by 48h till Aug 8, 11.59pm AoE.

Best regards,
  NeurIPS Area Chair

---

### Note · Authors · 2025-08-13

First, we sincerely thank each reviewer and the area chair for their valuable comments and efforts.

For Reviewers zqGh,4txP and mbYz, we greatly appreciate your constructive suggestions. We have engaged in thorough discussions, conducted extensive experiments and theoretical analyses to address your concerns, and further refined the paper based on your feedback. These improvements have made the work more complete.

For Reviewer e4GK, we are grateful for your insightful suggestions, and we regret that time constraints prevent us from engaging in further discussion. We have conducted comprehensive experiments to validate the universality of our self-distillation framework, showing that it remains effective in both purely spiking SNNs and ANN architectures. We have also clarified our choice of rate coding and how we address pathway inconsistencies between ANN and SNN.

Regarding your concern that the ANN-SNN structure deviates from the spike-firing principles in SNNs, we acknowledge this limitation but emphasize that it does not undermine the significance of our work, as outlined in our second-round response. First, due to resource constraints, we opted for a rate-coding-based approach to minimize costs; this approach is equivalent to spike firing for most static datasets. Moreover, as noted above, our self-distillation framework can also be applied to spike-firing-based SNNs, demonstrating its generality across different model structures. Second, from the perspective of ANN-to-SNN conversion, this approach aims to circumvent the non-differentiability of spikes in direct training by mapping pre-trained ANNs to SNNs. Since ANNs are inherently static and lack temporal structure, the resulting SNNs, while spike-based in form, often still convey information in a manner close to rate coding. Although this method does not explicitly encode temporal dynamics through precise spike timing, it remains an important and influential direction in SNN research due to its training efficiency and scalability, thereby indirectly supporting the relevance of our work.

Finally, we once again thank all reviewers and the area chair for their dedicated efforts.

---

### Decision · Program_Chairs · 2025-09-17

**Decision:**

Accept (poster)

**Comment:**

The authors propose an enhanced self-distillation framework for efficient training of Spiking Neural Networks (SNNs). Reduced training complexity and high performance are demonstrated in experiments on standard visual benchmark tasks.
Strengths:
- A novel and innovative method for efficient SNN training is proposed
- It shows improved training efficiency
- Good empirical results are obtained
- Includes a theoretical analysis of effects of reliable and unreliable teacher signals
Weaknesses:
- The method relies on rate-based approximations, does not exploit temporal dynamics
- The method is based on well-known approaches, which somewhat limits its novelty.
In conclusion, the manuscript presents interesting analyses and very good performance. The method is innovative within the SNN field. There are some weaknesses, but in summary the contribution was well received by the reviewers. I therefore propose acceptance.